# Habitat selection by wolves and mountain lions during summer in western Montana

**Collin J. Peterson** [1¤]*, **Michael S. Mitchell** [2‡], **Nicholas J. DeCesare** [3‡], **Chad J. Bishop** [1‡], **Sarah S. Sells** [1‡]

**1** Montana Cooperative Wildlife Research Unit, Wildlife Biology Program, University of Montana, Missoula, MT, United States of America, **2** US Geological Survey, Montana Cooperative Research Unit, University of Montana, Missoula, MT, United States of America, **3** Montana Department of Fish, Wildlife, and Parks, Missoula, MT, United States of America

¤ Current address: Montana Department of Fish, Wildlife, and Parks, Kalispell, MT, United States of America
‡ These authors also contributed equally to this work.
* Collin.peterson@mt.gov

**Data Availability Statement:** All relevant data are within the manuscript and its Supporting Information files.

**Funding:** C. Peterson received funding from the USFWS Federal Aid in Wildlife Restoration grant

## Abstract

In the Northern Rockies of the United States, predators like wolves (*Canis lupus*) and mountain lions (*Puma concolor*) have been implicated in fluctuations or declines in populations of game species like elk (*Cervus canadensis*) and mule deer (*Odocoileus hemionus*). In particular, local distributions of these predators may affect ungulate behavior, use of space, and dynamics. Our goal was to develop generalizable predictions of habitat selection by wolves and mountain lions across western Montana. We hypothesized both predator species would select habitat that maximized their chances of encountering and killing ungulates and that minimized their chances of encountering humans. We assessed habitat selection by these predators during summer using within-home range (3rd order) resource selection functions (RSFs) in multiple study areas throughout western Montana, and tested how generalizable RSF predictions were by applying them to out-of-sample telemetry data from separate study areas. Selection for vegetation cover-types varied substantially among wolves in different study areas. Nonetheless, our predictions of 3rd order selection by wolves were highly generalizable across different study areas. Wolves consistently selected simple topography where ungulate prey may be more susceptible to their cursorial hunting mode. Topographic features may serve as better proxies of predation risk by wolves than vegetation cover-types. Predictions of mountain lion distribution were less generalizable. Use of rugged terrain by mountain lions varied across ecosystem-types, likely because mountain lions targeted the habitats of different prey species in each study area. Our findings suggest that features that facilitate the hunting mode of a predator (i.e. simple topography for cursorial predators and hiding cover for stalking predators) may be more generalizable predictors of their habitat selection than features associated with local prey densities.

#F16AF01202 (https://www.fws.gov/laws/lawsdigest/FAWILD.HTML), The Allen Foundation (https://www.allenfoundation.org/), and the University of Montana (www.umt.edu). The funders had no role in study design, data collection and analysis, decision to publish, or preparation of the manuscript.

**Competing interests:** The authors have declared that no competing interests exist.

## Introduction

Predators affect ecosystems directly by killing prey, and indirectly by influencing prey behavior and distribution [1,2]. Predation risk is the consequence of numerous ecological processes, including the density, behavioral state, and habitat selection tendencies of predators at multiple spatial scales, each of which may correspond with unique effects on the behavior, distribution, and demography of prey. Fine scale selection (e.g. 3$^{rd}$ order or within-home range selection [3]) of habitat used for hunting by predators imposes varying levels of risk towards prey across space [4], and the perception of risk by prey can influence foraging behavior [5], and in turn alter vegetation communities [6,7]. Understanding how predators select habitat can yield predictions of space use [8], with implications for prey behavior, predator-prey interactions, and trophic dynamics [9].

The habitat of predators is conceptually akin to their realized niche, i.e., the resources and limiting factors required for population growth in the presence of competitors [10]. Depicting habitat according to niche-based components should also yield spatially-explicit predictions of species space-use that are generalizable across a wide range of ecological conditions [11]. The validity of such estimates can be assessed by testing how well predictions of predator space-use perform in multiple ecosystem-types. Moreover, generalizable predictions of habitat selection can obviate the need to conduct new studies of species distributions in other areas [12].

Factors directly regulating fitness, like food and the risk of encountering humans, should be important ultimate factors in shaping predator space-use [13,14]; however, when such direct measures are unavailable, proxies that correlate with the probability of encountering or capturing prey, like vegetation cover-types and topographic features, may also be useful for predicting local predator distributions. For example, open, less-rugged terrain may signify areas where cursorial predators like wolves (*Canis lupus*) can maximize opportunities to detect prey [15,16], and topographical features like drainages may enhance prey capture [17]. Dense vegetation and rugged terrain may serve as proxies for hiding cover that increases hunting success for ambush predators like mountain lions *(Puma concolor)* [18–21]. However, researchers studying ungulate behavior often assume certain vegetation cover-types are accurate proxies of predator distribution, without directly testing the relationship between those proxies and predator behavior [22–25]. Whereas ultimate factors driving a predator species' distribution may be consistent across broad scales, generalizable predictions at local scales may be difficult due to variation in important proximate habitat features, such as the composition of prey communities [12,13]. Hypothesizing *a priori* how proxies ultimately tie to the distribution of a species, then testing those hypotheses against location data from multiple populations, can increase generality of predictions of animal space-use [26].

Since the mid-1990s, wolves and mountain lions have increased in abundance and expanded their range within the Northern Rockies of the United States [27–31]. Summer is a critical period in the annual life-history of ungulates in the Northern Rockies, as the availability of forage during summer often regulates ungulate population growth rate [32], but forage acquisition by ungulates may be limited by wolf and mountain lion predation risk [33,34]. However, fine-scale approximations of space-use by these predators and an understanding of their effects on ungulate behavior are lacking in many parts of the Northern Rockies, prompting the need for generalizable predictions of wolf and mountain lion habitat selection in the region [20,35,36].

Wolves are group-living, territorial carnivores that primarily prey on elk (*Cervus canadensis*), mule deer (*Odocoileus hemionus*), white-tailed deer (*O. virginianus*), and moose (*Alces alces*) in the Northern Rockies [37,38]. As cursorial hunters, wolves frequently select topographically simple terrain like drainage bottoms, where they can travel quickly and pursue

prey over long distances [17,39,40]. Wolves will often select open vegetation cover-types that contain high quality forage for ungulates to increase encounter rates and more easily chase down prey [2,14]. For every pursuit wolves engage in, they have a relatively low probability of capturing prey [40], thus, selecting habitat that maximizes encounters with prey increases the hunting success of wolves [39].Wolf behavior is also shaped by human encounter risk, especially within hunted populations. Roads can increase risk of encountering hunters, trappers, and vehicles, but may also serve as energetically efficient travel routes while hunting, allowing wolves to travel farther and faster and increase their encounter rate with prey [41,42]. Wolves may respond differently to roads at different road densities (i.e., a functional response in selection [43–45]), reflecting a context-dependent tradeoff between the risk of encountering humans and the benefits of efficient travel and access to prey [46]. Thus, it is particularly important to investigate functional responses in selection for roads by wolves to understand how they attenuate exposure to risk [44].

Mountain lions are typically solitary, stalking predators. As the most widely distributed land mammal (besides humans) in the western hemisphere, mountain lions are prey generalists and exhibit a high degree of behavioral plasticity across biomes [45]. In the Northern Rockies, mountain lions primarily prey on elk, mule deer, white-tailed deer, and bighorn sheep (*Ovis canadensis*) [39,47,48]. Mountain lions are unlikely to make a kill if they begin an ambush >25 m away [18], so rely on hiding cover for hunting. They often select structurally complex, rugged topography that provides fine-scale hiding cover like boulders and outcrops [48], but will also select dense vegetation cover-types like thick forests and riparian areas within topographically simple areas [19–21,49]. Thus, mountain lions may exhibit weaker selection for features associated with prey encounter than wolves [18,39]. Mountain lion responses to roads are highly variable across ecosystems [49,50].

Our goals were to understand and predict 3rd order habitat selection by wolves and mountain lions during summer across a spectrum of ecological conditions in western Montana. Summer is a critical period in the life history of ungulates [32], where tradeoffs between nutrition and risk can influence their dynamics [33,34], so we were interested in characterizing predation risk during this period. We hypothesized that wolves and mountain lions select habitat that maximizes chances of encountering and killing ungulates and minimizes chances of encountering humans. We predicted wolves select drainage bottoms and low slopes as kill rates of ungulate prey have been shown to increase in these areas [17,40], and select areas with higher road densities to facilitate quick travel while hunting [51] and maximize encounters with prey [41,42].We also predicted that wolves select open-canopy vegetation cover-types because these areas are often selected by elk and deer [52,53]. We predicted mountain lions select hiding cover in the form of high canopy forests, forested drainage bottoms, and rugged terrain that can facilitate stalking prey, and that they avoid open vegetation cover classes. We predicted mountain lions avoid roads to reduce chances of encountering humans (Table 1).

To predict habitat selection by wolves and mountain lions in western Montana, we developed resource selection functions (RSFs) using Global Positioning System (GPS) collared animals. We developed separate, study area-specific RSFs for wolves and mountain lions in multiple study areas (Fig 1) that varied in prey community composition, dominant vegetation cover-types, and topographic complexity. To assess the generality of our RSFs, we applied each study area-specific RSF to out-of-sample GPS telemetry data from separate study areas for wolves and assessed their predictive performance. Very high frequency (VHF) telemetry datasets were the only out-of-sample telemetry data available for mountain lions in northwest Montana [20], and consisted of low sample sizes relative to GPS datasets. This precluded us from assessing the generality of more detailed models like step selection functions because they require animal locations sampled at finer temporal scales than standard RSFs [62].

**Table 1. Variables tested in resource selection functions with hypothesized biological relevance and predicted effect on wolf and mountain lion habitat selection.**

| Variable | Prediction | | Hypothesis | | Reference | Data source |
|---|---|---|---|---|---|---|
| | Wolves | Lions | Wolves | Lions | | |
| Canopy cover | - | + | Select low canopy cover to increase encounters with prey | Select high canopy cover for stalking prey | [16,21] | [54] |
| Road density | + | - | Select high road densities for energetic efficiency and to increase prey encounters | Avoid high road densities to avoid humans | [41,44,49,55] | [56] |
| Road * canopy cover interaction | + | + | Selection for roads will be stronger in areas with high canopy cover due to proximity to concealment | | | [54,56] |
| Slope | - | - | Select low slopes for easier travel and to chase down and kill prey | | [13] | 'terrain' function from 'raster' package in R [57] |
| Topographic position index [TPI] | - | 0 | Select drainage bottoms to chase down and kill prey | Indifferent to drainage bottoms | [17,58,59] | 'terrain' function from 'raster' package in R [57] |
| Terrain ruggedness index [TRI] | - | + | Select low ruggedness for chasing down prey | Select high ruggedness for stalking prey | [58,59] | 'terrain' function from 'raster' package in R [57] |
| Burns | + | 0 | Select burns to encounter prey | Indifferent to burns | [16,21] | [60,61] |
| Grass & shrublands | + | 0 | Select grass & shrublands to encounter prey | Indifferent to grasslands | [16,21] | [61] |
| Recently logged forests | + | 0 | Select harvests to encounter prey | Indifferent to harvests | [16,21] | [60,61] |

"+" indicates predicted selection for high values of a variable, "-" indicates avoidance and "0" indicates neutrality. Some variables were not tested in both wolf and mountain lion RSFs due to collinearity with other variables.

However, VHF data enabled us to test the generality of mountain lion RSFs, and we used RSFs for both mountain lions and wolves to keep our methods consistent between species.

## Study areas

We studied wolf and mountain lion habitat selection within 6 different study areas throughout western Montana, USA: 3 wolf study areas, and 3 mountain lion study areas, some of which overlapped substantially (Fig 1).

### The Garnet Range mountain lion study area

The Garnet Range study area (48.83˚N, -113.66˚W) encompassed 2,840 km². Elevations ranged from 1,160–2,156 m (Fig 1). The Garnet Range were characterized by relatively moderate rolling topography, and primarily consisted of mesic forests and recently logged forests. From 2001–2006, Garnet Range mountain lions were sympatric with black bears (*Ursus americanus*), grizzly bears (*U. arctos*), coyotes (*C. latrans*), bobcats (*Lynx rufus*), and Canada lynx (*L. canadensis*) [63]. One wolf pack was verified in the Garnet Range in 2006 [63]. The ungulate community in the Garnet Range was composed of white-tailed deer, mule deer, elk, and moose.

### The Rocky Mountain Front wolf and mountain lion study areas

The study areas for wolves and mountain lions on the Rocky Mountain Front, MT, USA (47.44˚N, -113.12˚W and 47.61˚N, -112.75˚W, respectively) encompassed 3,605 km² and 1,223 km², respectively, with elevations ranging 1,240–2,800 m. The wolf study area almost fully encompassed the mountain lion study area in that region, stretching farther west, south, and east than the mountain lion study area, and included more of the Bob Marshall Wilderness Complex (Fig 1). This region represented the transition zone between the Great Plains and the Rocky Mountains and contained a pronounced east-to-west gradient in dominant vegetation cover classes, elevation, and topographic complexity. The eastern portion of the Rocky

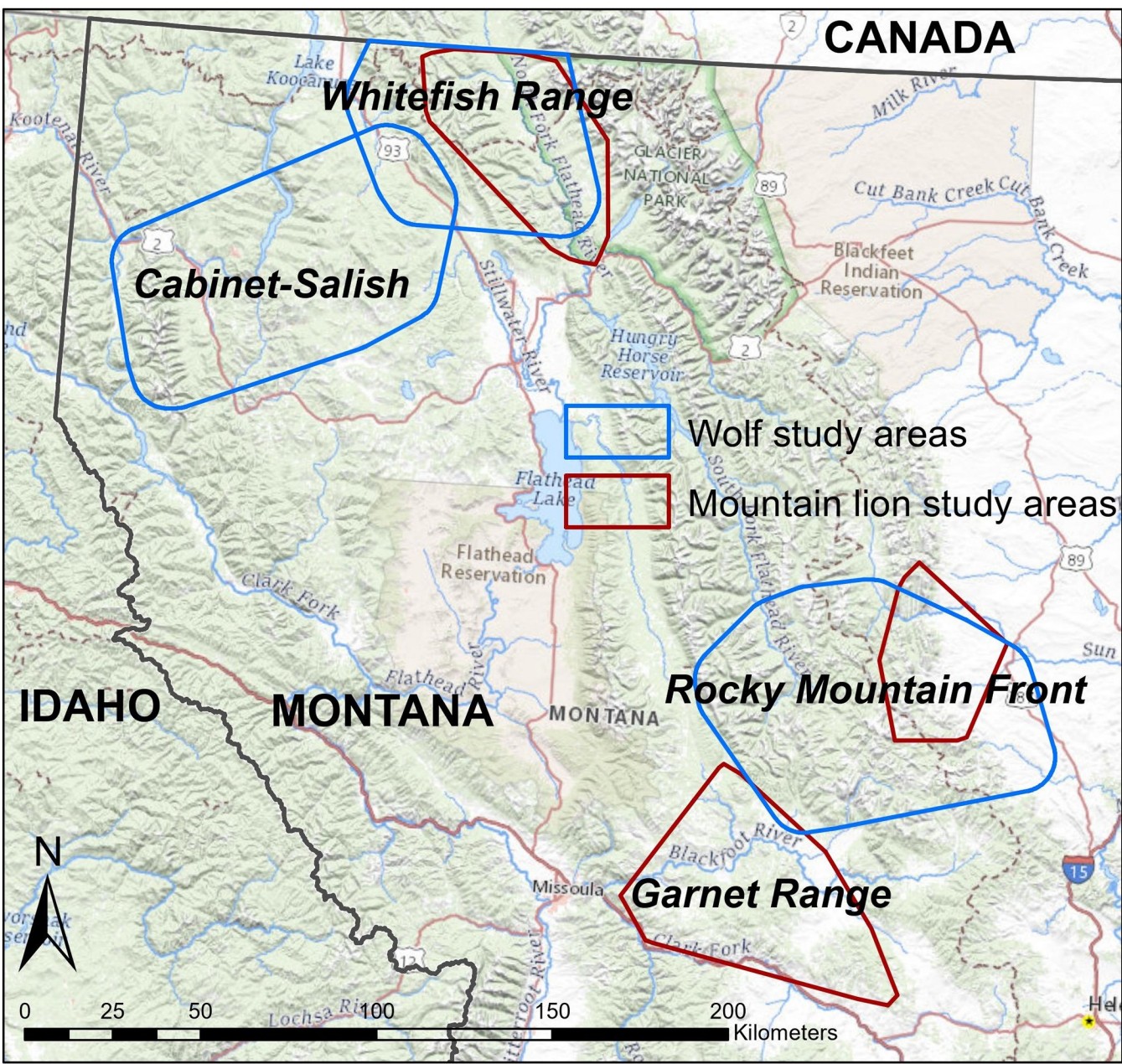

**Fig 1. Wolf and mountain lion study areas.** Polygons are 100% minimum convex polygons (MCPs) surrounding wolf (in blue) or mountain lion (in red) Global Positioning System- (GPS) or Very High Frequency- (VHF) collar locations during summer that were used for resource selection function (RSF) modelling and testing. Base map was reprinted from the U.S. Geological Survey National Map and is without copyright.

Mountain Front comprised open, relatively flat mixed grass prairie, and the western portion contained the mountains of the Bob Marshall Wilderness, characterized by steep, cliffy terrain and a patchy mosaic of burned areas, mesic conifer forests, meadows, and subalpine steppe. The ungulate community included mule deer, white-tailed deer, elk, moose, bighorn sheep, mountain goats (*Oreamnos americanus*), and pronghorn (*Antilocapra americana*). Carnivores included wolves, mountain lions, coyotes, grizzly bears, black bears, bobcats, and Canada lynx.

## Whitefish Range wolf and mountain lion study areas

The study areas for wolves and mountain lions in the Whitefish Range, MT, USA (48.88˚N, -114.73˚W and 48.81˚N, -114.33˚W, respectively) encompassed 2,832 km², and 2,065 km², respectively, with elevations of 780–2,400 m. The mountains were dominated by wet and mesic conifer forests, and a smaller proportion of subalpine forest, open grasslands, burned areas, and recently logged forests. The study area was bordered by Canada and Glacier National Park. The Whitefish Range was home to the same carnivore and ungulate species as the Rocky Mountain Front, except mountain goats and pronghorn.

## Cabinet-Salish wolf study area

The Cabinet-Salish study area (48.28˚N, -115.31˚W) encompassed 2,832 km², with elevations of 630–2,700 m. The study area was bisected by the Fisher River. The Salish Mountains to the east were characterized by moderate, rolling topography, and mesic forests, grasslands, and recently logged forest. The Cabinet Mountains to the west were relatively steeper and more rugged, with wet forest transitioning to subalpine areas. The Cabinet-Salish was home to the same carnivores and ungulate species as the Rocky Mountain Front, except pronghorn.

## Methods

### Data sources and animal captures and handling

To study habitat selection by wolves, we used GPS-collar data from packs in the Cabinet-Salish, the Rocky Mountain Front, and the Whitefish Range, MT. During 2015–2018, wolf specialists with Montana Fish Wildlife and Parks (MFWP) captured wolves using summer ground and winter aerial captures. Ground captures were conducted with foothold traps designed with offset teeth and rubber-coated jaws to reduce injury (EZ Grip # 7 double long spring traps, Livestock Protection Company, Alpine TX). Aerial captures were conducted by MFWP-contracted crews using helicopters and dart guns. Wolves were anesthetized and handled in accordance with MFWP's biomedical protocol for free-ranging wolves [64], guidelines from the Institutional Animal Care and Use Committee for the University of Montana (AUP # 070–17) and guidelines approved by the American Society of Mammologists [65]. Wolves were fit with GPS collars (models Lotek LifeCycle, Lotek Litetrack B 420, Telonics TGW-4400-3, Telonics TGW-4483-3, or Telonics TGW-4577-4) that collected fixes every 4–13 hours (depending on collar model). In total, our dataset consisted of GPS telemetry data from 18 individual wolves among 13 packs, containing 1,019 locations from 4 packs in Cabinet-Salish, 2,698 locations from 6 packs on the Rocky Mountain Front, and 815 locations from 3 packs in the Whitefish Range.

To study habitat selection by mountain lions, we used radiotelemetry data from collared mountain lions in 3 study areas: The Garnet Range, the Whitefish Range, and the Rocky Mountain Front (Fig 1). The Garnet Range data consisted of 40,831 GPS collar locations collected during summers of 2001–2006 from 17 mountain lions (14 females, 3 males) that were collared as part of a previous long-term study by Robinson and DeSimone [63]. Mountain lion GPS collars collected fixes every 5 hours. The Whitefish Range data consisted of 875 VHF radiotelemetry collar locations from 34 mountain lions (25 females, 9 males) collected during summers 1992–1996 in a study by Kunkel et al. [59]. Kunkel et al. relocated mountain lions from the ground opportunistically on a daily basis, and collected locations of all animals weekly from the air. The Rocky Mountain Front data consisted of 145 VHF telemetry locations from 20 mountain lions (12 females, 8 males) collected during summers 1991–1992 in a study by Williams [66]. Williams relocated mountain lions 2 times per month from the air and

infrequently from the ground, and estimated relocation accuracy to be within 150–200 m. Capture and handling information for collared mountain lions are detailed in existing publications [59,63,66]. Of these studies, Williams' [66] was the only one that characterized habitat-use by mountain lions during the summer, and none of these studies assessed resource selection via a used-available design, thus, our research offers new insights into resource selection by mountain lions during summer in western Montana.

### Developing wolf RSFs

We developed 3rd order (within-home range) summer RSFs [3] for wolves using GPS-collar locations collected between June 1 and September 1, 2015–2018. In an effort to remove locations collected while wolves were likely not travelling (and therefore not hunting [17]), we calculated step lengths and movement rates between subsequent locations of each wolf, and removed locations preceding steps with movement rates of <0.025 km/hr (i.e. step lengths <100 m in 4 hr) using the 'amt' package [67] in R version 3.5.1 [68]. This is an important procedure for focusing on hunting behavior of wolves, because wolves spend much of their time relatively stationary at rendezvous sites during the summer [35,69], and also helps reduce spatial autocorrelation. This omitted 1,821 locations, leaving 2,711 locations for developing the RSFs. We used these data to construct 95% kernel density estimate (KDE) home ranges for each individual wolf using the adehabitatHR package [70] in R, with 'href' as the smoothing parameter [71]. For used samples, we included GPS locations that were within individual home ranges. For available samples, we randomly sampled 5 points per used location within that individual's home range [71] totaling 13,610 available locations.

We tested the effects of variables that have previously been shown to correlate with search and kill rates and risk of human encounter for wolves in RSFs. These included topographic position index ([TPI] a continuous metric of landform category that distinguishes features like ridgelines from valley bottoms by comparing the elevation of a cell in a digital elevation model to the mean elevation of a 1 km$^2$ window around that cell), terrain ruggedness index ([TRI] a metric of ruggedness, calculated as the mean of the absolute differences between elevation at a cell and the 8 surrounding cells of a 30m$^2$ digital elevation model), slope, vegetation cover-type, forest canopy cover, and road density variables (Table 1). To classify vegetation cover-types, we reclassified a state landcover map developed by the Montana Natural Heritage Program (MNHP) [61] into vegetation cover-classes of riparian, conifer forest, grasslands-shrublands, recently logged forests, burns, and other (cover-types that did not fall into these categories and composed <2% of the study area). This map was intended to reflect conditions in 2017, which overlaps with the timing of wolf GPS collar locations (but does not overlap with mountain lion telemetry studies). The map was classified from 30-m resolution satellite and aerial imagery from the year 2000, was field-validated in 2008, and was updated with current imagery in 2017 and is the most accurate landcover dataset available for Montana over the past 2 decades [C. Tobalske, MNHP, personal communication]. We identified additional burns and recently-logged forests present in the Whitefish Range, Rocky Mountain Front, and Cabinet-Salish between 2015–2018 (while wolf data were collected) using disturbance maps from MFWP and LANDFIRE [60]. To quantify road density, we used a statewide transportation infrastructure map [56] to calculate the total length of paved and unpaved roads (km) per km$^2$ in each study area. We assumed all spatial covariates were measured without error, resulting in RSF coefficient estimates being more precise than they would have been had we properly accounted for uncertainty in covariate values.

We used logistic regression to develop 4 RSFs for wolves: 3 fixed-effects models (developed separately for the Cabinet-Salish, the Whitefish Range, and the Rocky Mountain Front) and 1

mixed-effects RSF for all study areas combined. We calculated Pearson's correlation coefficients (*r*) and did not include variables with *r* > 0.5 in the same model. For collinear variables, we retained the variable that yielded lowest Akaike's Information Criteria (AIC) in a single variable RSF for a given species and study area. For each study area, we developed a global model that included the full suite of variables and biologically interpretable two-way interactions. Conifer forest was the reference vegetation-cover type category in global models, since it was the most abundant cover type available within wolf and mountain lion home ranges. Continuous variables were centered and scaled. We screened each global model for uninformative parameters by ranking each variable by level of importance (estimated as the absolute value of β/SE), then sequentially removed one variable at a time in ascending order of importance [72,73]. If removal of a variable reduced AIC values, it was discarded from the model. If removal of a main effect decreased AIC, but removal of that main effect in an interaction increased AIC, the main effect and interaction were retained. We repeated this process until no additional variable could be removed without increasing AIC. To further control for multicollinearity, we calculated variance inflation factors (VIFs) for each variable and eliminated variables with VIF>5. We calculated 95% confidence intervals (CI's) on the coefficients for each variable and eliminated variables whose CI overlapped zero from final models, unless the CI of a main effect overlapped zero but the CI of an interaction with that main effect did not.

In balancing multiple objectives of predicting wolf behavior across multiple ecosystems and understanding the drivers of variation in wolf behavior, we focused the bulk of our analyses on developing study area-specific fixed-effects models, which are simple for making spatially-explicit predictions of resource selection. We used mixed-effects logistic regression specifically to test whether selection for density of roads by wolves changed as a function of road availability in each pack's territory. We combined GPS data from our 3 study areas for wolves and determined a top fixed-effects RSF for all packs using the same model selection procedures outlined above [73]. We then added a random intercept for each pack and a random coefficient for the effect of roads to this model, and calculated pack-specific slope coefficients for the effect of roads on selection [74]. We converted coefficients to the relative odds scale, plotted the trend line between relative odds of selection for roads and mean road density in each pack's territory, and calculated the slope of this function, where a non-zero slope provided evidence for a functional response in selection of roads by wolves [75].

## Developing mountain lion RSFs

VHF telemetry datasets for mountain lions on the Rocky Mountain Front and Whitefish Range were of insufficient size for modeling habitat selection directly, so we used data from the Garnet Range to achieve 2 major goals: (1) develop an RSF using data from the Garnet Range that best explained mountain lion habitat selection within that study area, and (2) develop RSFs using data from the Garnet Range that best predicted mountain lion distribution in the Whitefish Range and Rocky Mountain Front. We developed 3rd order fixed-effect RSFs for mountain lions in the Garnet Range, and included all locations because 3rd order mountain lion habitat selection does not vary substantially when mobile versus stationary [21,76], and mountain lions do not concentrate activity at rendezvous sites like wolves do. We constructed 99% KDEs to estimate mountain lion home ranges, because 95% KDEs (used for wolves) resulted in fragmented home ranges that appeared to omit important travel corridors [77]. We used the same method for sampling used and available locations for mountain lions as for wolves, totaling 40,831 used locations and 204,155 available locations for mountain lions in the Garnet Range. We tested variables that have previously been shown to correlate with search and kill rates and risk of human encounter for mountain lions (Table 1) to develop an

RSF that best explained mountain lion habitat selection within the Garnet Range. We used the MNHP vegetation cover-type map from 2017 to classify cover-types in the Garnet Range with the same approach we used for wolves, and identified additional burns present within 2001–2006 using disturbance maps from MFWP and LANDFIRE. We categorized recently logged forests as conifer forest in the Garnet Range, since we lacked timber harvest layers from this region.

To better understand how habitat use differed between mountain lions on the Rocky Mountain Front and Whitefish Range (so that RSFs could be catered to better predict habitat selection in those areas), we first plotted VHF telemetry locations from those study areas on a topographic map. We initially observed that mountain lions in the Whitefish Range used less rugged terrain than on the Rocky Mountain Front visually, and confirmed this by comparing mean TRI at used locations of mountain lions in all study areas. So, we developed a "moderate ruggedness" model that included a quadratic effect of TRI for predicting habitat selection in the Whitefish Range, and a "high ruggedness" model that omitted this quadratic effect for predicting habitat selection on the Rocky Mountain Front.

### Testing generality of RSF predictions

For wolves and mountain lions, we tested the validity of fixed-effect RSFs internally through $k$-folds cross validation, and tested their generality in separate ecosystem-types by applying each study-area-specific RSF to out-of-sample data from the other 2 study areas, respectively [78]. For internal validation, we partitioned each dataset used to build study area-specific RSFs into 5 folds and calculated an RSF value for each used location in each fold using the RSF developed from the original dataset. We divided predicted RSF values into 10 bins separated at equal intervals, and assigned each predicted RSF value a rank ranging from 1 to 10. We determined area-adjusted frequency of use of each bin by dividing the frequency of cross-validated used locations within a bin by the area of that range of bin scores available in a study area. Then, we calculated a Spearman's rank correlation ($r_s$) between area-adjusted frequency of use and RSF bin-rank.

For external validation, we applied each study area-specific RSF for wolves to the used location datasets of the remaining 2 wolf study areas (not divided into folds). We binned RSF predictions into 10 equal-area deciles and calculated $r_s$ between decile bin-rank and frequency of use within each binned RSF decile. For mountain lions, we tested how well the moderate and high ruggedness models predicted the relative probability of selection at locations from VHF-collared mountain lions on the Rocky Mountain Front in the Whitefish Range. We binned these predicted RSF values into 10 equal-area deciles and calculated $r_s$ between RSF decile bin-rank and frequency of use of each binned RSF decile.

## Results

### Wolf RSFs

Wolves selected drainage bottoms (low TPI values), and low slopes consistently across study areas (Table 2). Selection for vegetation cover-types varied by study area (Table 2). Relative to conifer forests, wolves avoided grass-shrublands in Cabinet-Salish but grass-shrublands had no effect in the other 2 wolf study areas. Wolves selected burns in the Whitefish Range and Rocky Mountain Front, but burns had no effect on selection in Cabinet-Salish. Wolves only selected recently logged forests on the Rocky Mountain Front. Selection for canopy cover and roads also varied by study area. On average, wolves selected high canopy cover on the Rocky Mountain Front and Whitefish Range, but slightly avoided high canopy cover in Cabinet-Salish (Table 2). Wolves generally avoided roads in Cabinet-Salish and on the Rocky Mountain

**Table 2. Resource selection coefficients and confidence intervals (CIs) from top-ranked fixed effects and mixed-effects resource selection functions for wolves.**

| Parameter | Cabinet-Salish | | | Rocky Mtn. Front | | | Whitefish Range | | | Global model | | |
|---|---|---|---|---|---|---|---|---|---|---|---|---|
| | β | 95% CI | | β | 95% CI | | β | 95% CI | | β | 95% CI | |
| | | 2.5% | 97.5% | | 2.5% | 97.5% | | 2.5% | 97.5% | | 2.5% | 97.5% |
| Canopy cover | -0.105 | -0.204 | -0.004 | 0.079 | 0.019 | 0.14 | 0.145 | 0.034 | 0.258 | 0.181 | 0.129 | 0.233 |
| Grass-shrublands | -0.395 | -0.763 | -0.048 | | | | | | | | | |
| Other landcovers | | | | 0.516 | 0.323 | 0.706 | | | | 0.257 | 0.117 | 0.396 |
| Grass-shrublands × TPI | 0.438 | 0.108 | 0.756 | | | | | | | | | |
| Recently logged | | | | 0.657 | 0.329 | 0.974 | | | | | | |
| Road density | -0.125 | -0.222 | -0.03 | -0.24 | -0.328 | -0.154 | 0.266 | 0.166 | 0.365 | -0.187 | -0.475 | 0.101 |
| Road density × canopy cover | | | | 0.127 | 0.037 | 0.216 | | | | -0.048 | -0.097 | 0.001 |
| TPI | -0.523 | -0.63 | -0.419 | -0.311 | -0.379 | -0.244 | -0.405 | -0.534 | -0.278 | -0.376 | -0.427 | -0.326 |
| Slope | -0.526 | -0.63 | -0.425 | -0.641 | -0.712 | -0.572 | -0.556 | -0.696 | -0.422 | -0.721 | -0.777 | -0.665 |
| Wildfire | | | | 0.411 | 0.285 | 0.538 | 0.299 | 0.017 | 0.575 | 0.742 | 0.638 | 0.847 |
| Random intercept of pack | | | | | | | | | | | Variance | |
| | | | | | | | | | | | 0.567 | |
| Random intercept road density×pack | | | | | | | | | | | Variance | |
| | | | | | | | | | | | 0.611 | |
| Random slope of road density×pack | | | | | | | | | | | Variance | |
| | | | | | | | | | | | 0.176 | |

Front, but in the latter study area there was a positive road × canopy cover interaction. When plotted, visual assessment of this interaction indicated wolves avoided roads weakly within high canopy cover areas (Fig 2). On average, wolves selected roads in the Whitefish Range. Our multi-study area mixed-effects model provided modest evidence that relative odds of selection for roads increased as mean road density in a pack's territory increased. The pooled mean effect of road density on selection across all packs was negative (β = -0.074, 95% CI = [-0.345–0.196]), but varied from negative to positive between individual packs (β = -0.751–0.273). For every 1 km per $km^2$ increase in road density within a pack's territory, the relative odds of selection for roads increased by 19.2% ($P$ = 0.098; 95% CI = [-4.1%–42.5%]; Fig 3).

## Wolf RSF generality

Study area-specific wolf RSFs performed well during internal model validation ($r_s$ = 0.957–0.967). Study area-specific RSFs were highly generalizable; mean $r_s$ from models developed in other study areas and applied to testing data also indicated excellent model fit ($r_s$ = 0.952–0.999; Fig 4).

## Mountain lion RSFs

The moderate ruggedness RSF best explained mountain lion habitat selection in the Garnet Range and included topographic position, road density, canopy cover, "other" landcover types, terrain ruggedness (TRI) and a quadratic effect of TRI (Table 3). Mountain lions in the Garnet Range selected drainage bottoms, but there was a negative interaction between topographic position and canopy cover. When plotted, visual assessment of this interaction indicated mountain lions selected ridgelines and peaks with low canopy cover (Fig 5). With all other variables held at their mean, mountain lions selected areas with higher canopy cover, and selected areas with moderate ruggedness, as indicated by the quadratic effect of TRI. Garnet Range mountain lions avoided roads. The high ruggedness RSF included the same

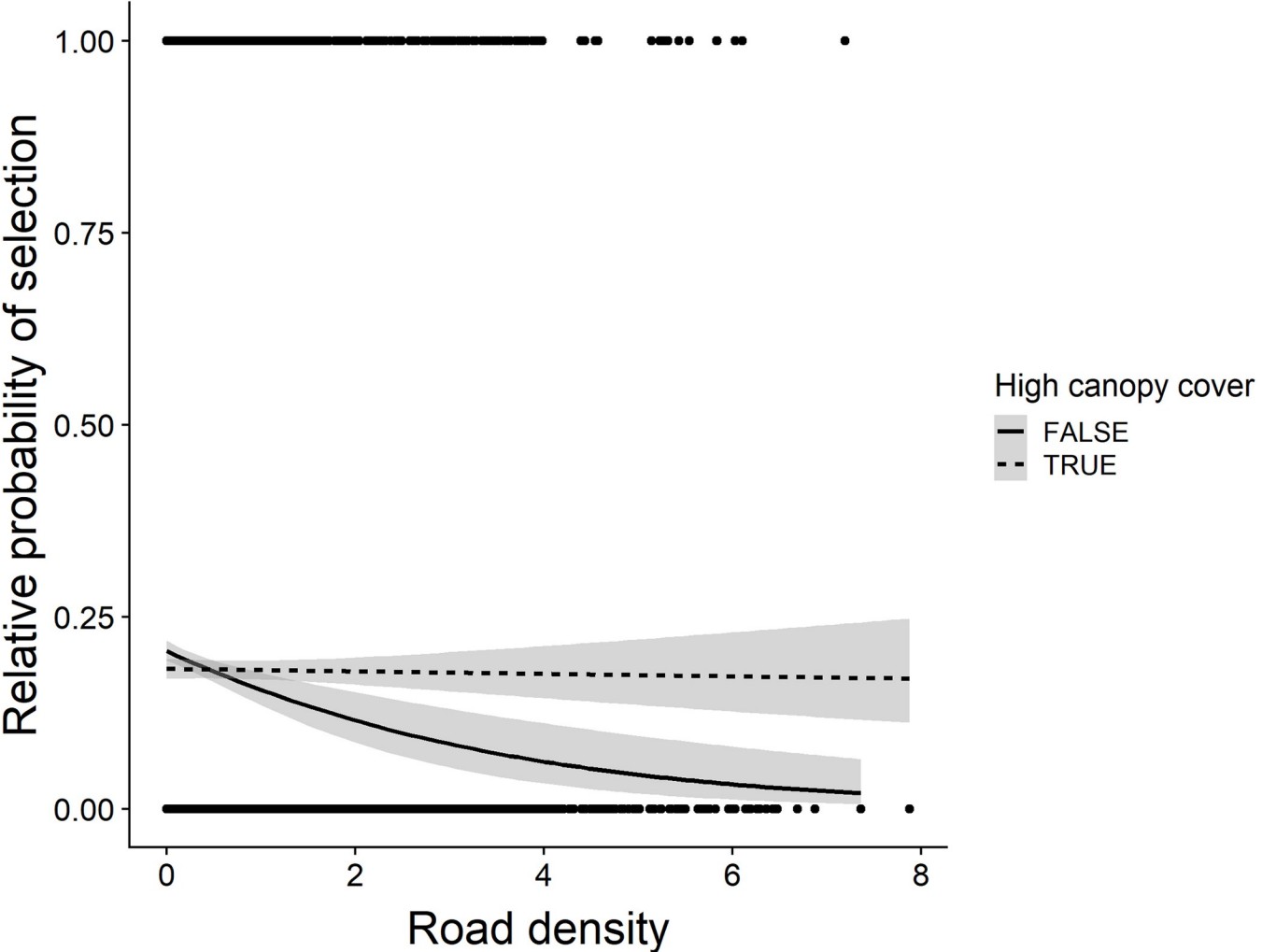

**Fig 2. Selection for road density in high versus low canopy cover by wolves on the Rocky Mountain Front.** Relative probability of selection for roads in high canopy cover (>35%, dashed line) versus low canopy cover (<35%, solid line) areas by wolves on the Rocky Mountain Front, MT, USA. This cutoff was calculated as the median value of canopy cover at available location.

covariates and similar parameter estimates as the moderate ruggedness RSF, however, it only included a positive effect of TRI, therefore predicted lions selected more rugged terrain.

### Mountain lion RSF generality

Both the moderate ruggedness and high ruggedness RSFs performed well during internal validation in the Garnet Range ($r_s$ = 0.952 and 0.939, respectively; Fig 6). Mountain lions on the Rocky Mountain Front used more rugged terrain than in the Whitefish and Garnet ranges. Mean TRI at locations used by mountain lions on the Rocky Mountain Front ($\bar{x}$ = 80.882, SD = 30.299) was 1.76 times higher than in the Whitefish Range ($\bar{x}$ = 45.85, SD = 43.266) and 1.22 times higher than in the Garnet Range ($\bar{x}$ = 66.213, SD = 27.244). Accordingly, the moderate ruggedness RSF performed poorly on the Rocky Mountain Front $r_s$ = -0.863; Fig 6), but the high ruggedness RSF performed fairly ($r_s$ = 0.673; Fig 6). In the Whitefish Range, the moderate ruggedness RSF performed well ($r_s$ = 0.936; Fig 6), but the high ruggedness RSF performed poorly ($r_s$ = -0.952; Fig 6).

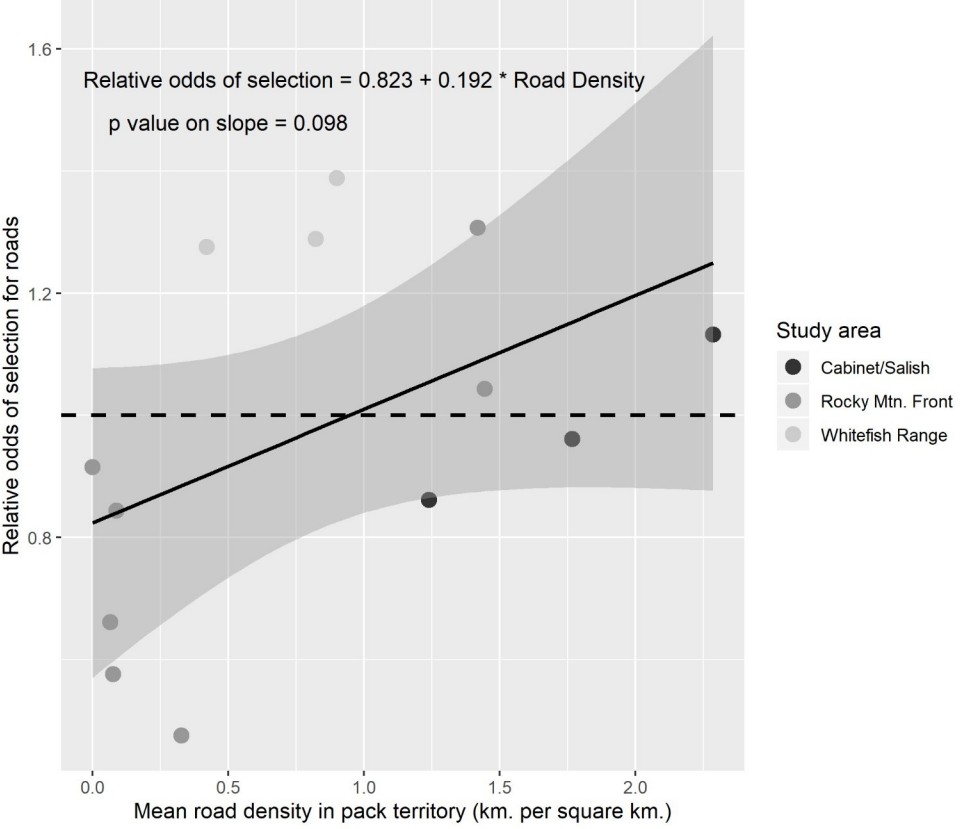

**Fig 3. Relative odds of selection for roads by wolves.** Relative odds of selection for road density by pack from mixed effects logistic regression model of within-territory resource selection by wolves, plotted against mean road density in each pack's territory. Dashed line at y = 1 indicates neutral selection.

## Discussion

We hypothesized that wolves and mountain lions would select habitat that increased their likelihood of encountering and killing ungulates, and that decreased their likelihood of encountering humans during summer. We found support for these hypotheses with wolves across multiple ecosystem-types. Due to their cursorial hunting mode and reliance on ungulate prey, we expected wolves to select simple topography where detection and kill rates of ungulates are higher. Within each of our study areas, low slopes and drainage bottoms predicted selection by wolves, supporting our hypothesis. To increase their chances of encountering ungulates, we also expected wolves to select open vegetation cover-types hypothesized to contain high quality forage. Our predictions were supported in the Whitefish Range and Rocky Mountain Front, where wolves selected open cover-types like burns and recently logged forests. However, wolves selected high canopy cover in those areas and avoided grass-shrublands in the Cabinet-Salish, lending mixed support to our hypothesis. For wolves, selection of vegetation cover-types associated with higher forage quality for ungulates was inconsistent across regions, suggesting that for ungulates, the risk of encountering wolves may be decoupled from forage dynamics. This may relieve elk and deer from having to make tradeoffs between forage and security from wolves during summer. Further, selection for vegetation cover-types by wolves may depend on the density of certain cover-types (e.g. timber harvests) in a region [44], and a functional response in selection of this nature could underlie the variance in selection for

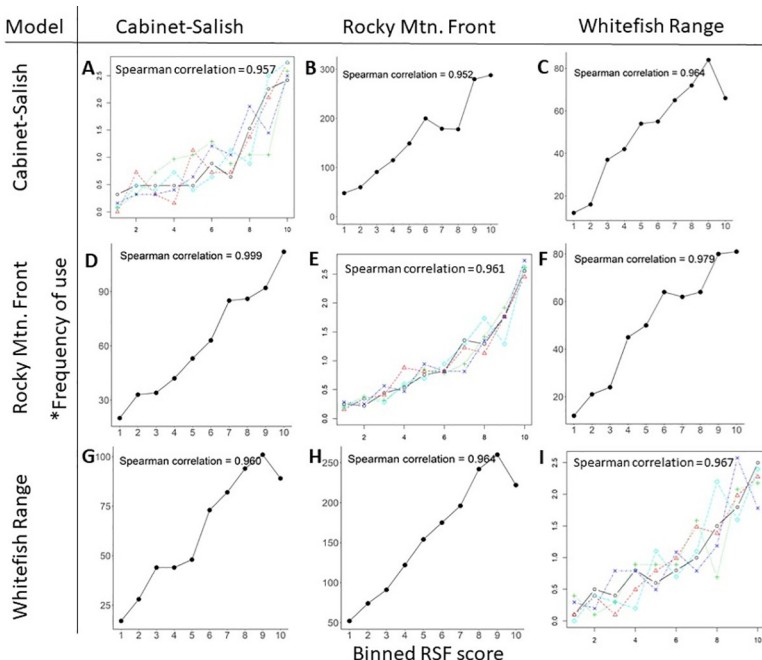

**Fig 4. Internal and external cross validation of wolf resource selection functions.** Rows indicate the study area in which data to develop each resource selection function (RSF) originated, and columns indicate the study area in which data to test each RSF originated. Frames 'A', 'E' and 'I' show results from internal k-folds cross validation, plotting area-adjusted frequency of use by wolves (y axis) per RSF bin (x axis) from fixed-effect logistic regression models developed with GPS collar data from wolves. Each line in these plots represents 1 out of 5 folds of data used to cross-validate RSF predictions. Mean Spearman correlations were calculated between each RSF bin rank and area-adjusted frequency of use in each bin. Frames 'B', 'C', 'D', 'F', 'G' and 'H' show results from external cross-validation with raw frequency of use (y axis) per equal-area binned RSF deciles (x axis). Spearman correlations were calculated between each RSF bin rank and frequency of use in each bin. *Note that area-adjusted frequency of use is on the y-axis for internal validation plots, whereas raw frequency of use is on the y-axis for external validation plots.

cover-types we observed. Nevertheless, each of our study area-specific RSFs for wolves was highly generalizable to different ecological conditions, indicating that selection for structurally simple topography is a common mechanism influencing local space use of wolves in western Montana. Given the consistent, strong effects of slope and topographic position index (TPI) across study areas, our results suggest that broad-scale avoidance of drainage bottoms and low slopes by ungulates may be more effective than avoiding open vegetation cover-types to circumvent predation risk from wolves in western Montana.

**Table 3. Resource selection coefficients and confidence intervals (CIs) from top-ranked fixed effects resource selection functions for mountain lions.**

| Parameter | Moderate ruggedness model | | | High ruggedness model | | |
|---|---|---|---|---|---|---|
| | β | 95% CI | | β | 95% CI | |
| | | 2.5% | 97.5% | | 2.5% | 97.5% |
| Canopy cover | 0.157 | 0.144 | 0.169 | 0.2 | 0.188 | 0.212 |
| Other landcovers | -0.303 | -0.351 | -0.256 | -0.29 | -0.337 | -0.243 |
| Road density | -0.067 | -0.078 | -0.055 | -0.069 | -0.08 | -0.057 |
| TPI | 0.035 | 0.023 | 0.047 | 0.073 | 0.061 | 0.085 |
| TPI × canopy cover | -0.206 | -0.218 | -0.195 | -0.234 | -0.245 | -0.223 |
| TRI | 0.282 | 0.267 | 0.297 | 0.148 | 0.136 | 0.16 |
| TRI$^2$ | -0.212 | -0.227 | -0.198 | | | |

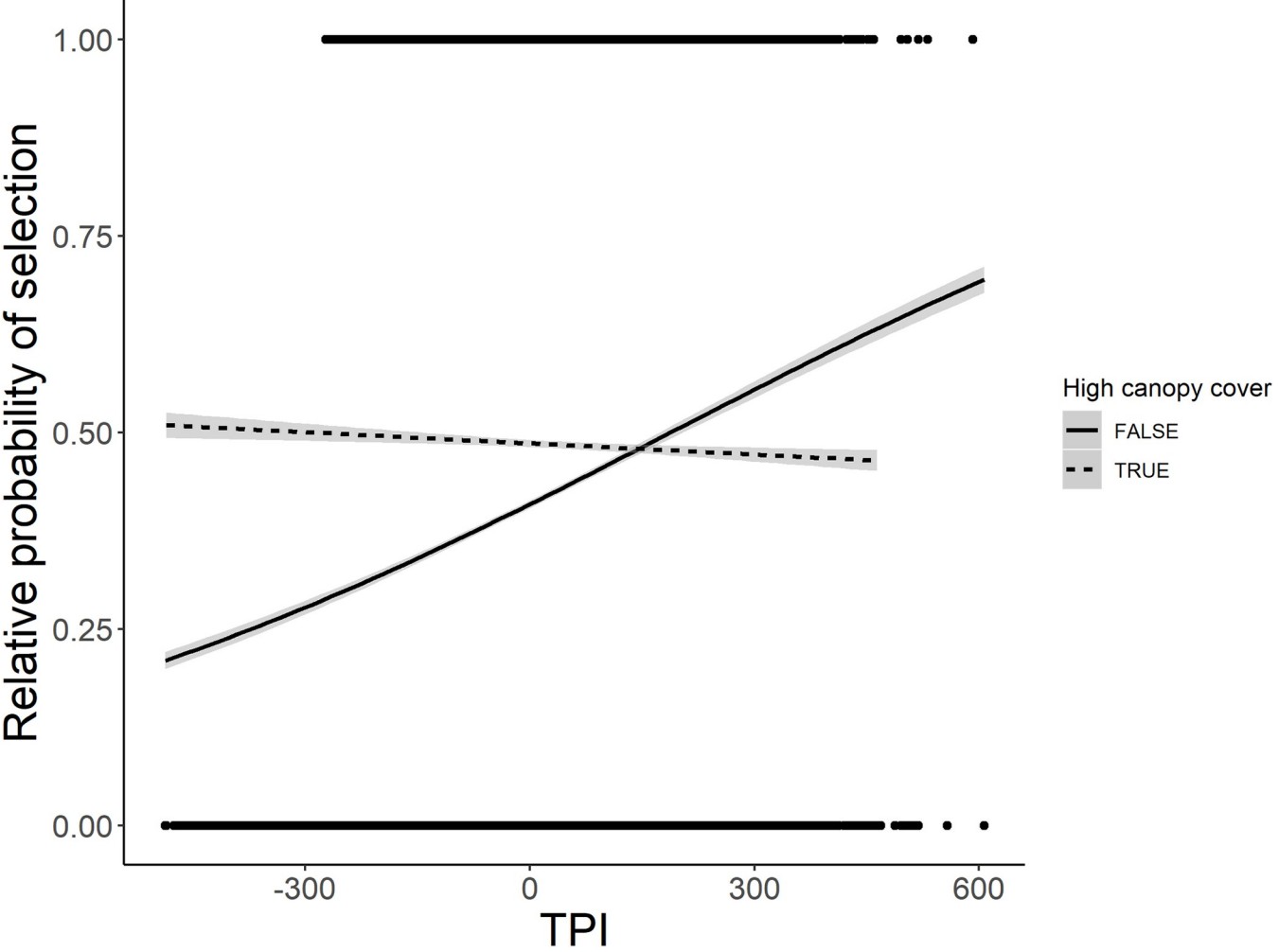

**Fig 5. Selection for topographic position in high versus low canopy cover by mountain lions in the Garnet Range.** Relative probability of selection for topographic position index (TPI) in high canopy cover (>54%, dashed line) versus low canopy cover (<54%, solid line) areas by mountain lions in the Garnet Range, MT, USA. This cutoff was calculated as the median value of canopy cover at available locations. A positive slope indicates selection for ridgelines and peaks, while a negative slope indicates selection for drainages.

Though wolves possess the capacity for dynamic habitat selection patterns within the same pack [46,79], their reliance on ungulate prey and the limitations of cursorial hunting resulted in predictable selection of simple topography across an array of ecosystem-types. Deer and elk compose the majority of wolf diets in western Montana [37,38], and during summer, these ungulates often seek refuge and forage in steep, high-elevation terrain [51,58,80], which could effectively reduce prey availability for wolves within simpler topography. However, for every pursuit wolves engage in, they have a low probability of capturing prey [40,81–83], so topography that makes prey more vulnerable is important for successful hunts [15,17]. Regardless of ecosystem-type, wolves likely selected drainage bottoms and low slopes due to increased prey vulnerability within those areas. It is important to note that these RSFs approximate the average, population-level summer behaviors of wolves, and do not account for the idiosyncratic hunting behaviors some individuals or packs can display. For example, certain wolf packs have been known to specialize on mountain goats [84,85], beavers (*Castor canadensis*) [86] and bison [82]. Though our RSFs showed that wolves primarily selected simple topography for

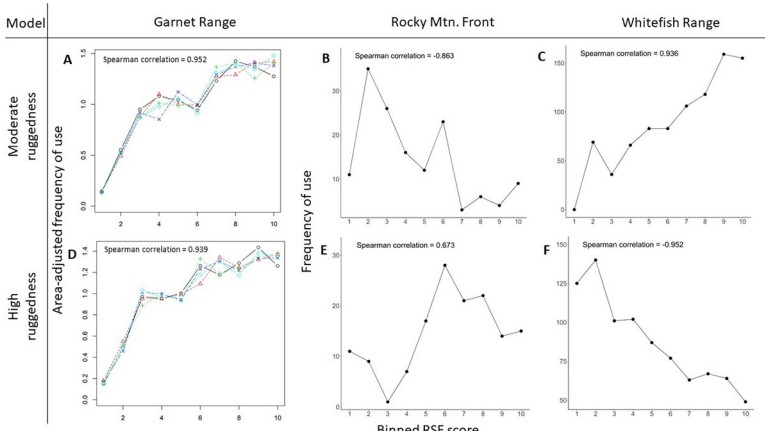

**Fig 6. External cross validation of mountain lion resource selection functions.** Rows correspond with different resource selection functions (RSFs) developed for mountain lions using GPS collar data from the Garnet Range, MT, USA, and columns indicate the study area in which data to test each RSF originated. Frames 'A' and 'D' show results from internal k-folds cross validation, plotting area-adjusted frequency of use by mountain lions (y axis) per RSF bin (x axis) from fixed-effect logistic regression models developed with GPS collar data from mountain lions. Each line in these plots represents 1 out of 5 folds of data used to cross-validate RSF predictions. Mean Spearman correlations were calculated between each RSF bin rank and area-adjusted frequency of use in each bin. Frames 'B', 'C', 'E' and 'F' show results from external cross-validation with raw frequency of use (y axis) per equal-area binned RSF deciles (x axis). Spearman correlations were calculated between each RSF bin rank and frequency of use in each bin.

capturing deer and elk, as generalists, wolves are certainly capable of exploiting a wider variety of terrain.

Our results were consistent with the hypothesis that wolves select habitat that minimizes chances of encountering humans. For wolves, roads pose a tradeoff between the risk of encountering humans, and the benefits of efficient travel and access to prey [46], and selection of roads often depends on the context of overall level of human use in their territory. In our study, wolves selected roads, but the strength of selection appeared dependent on the probability of being detected by humans. On the Rocky Mountain Front, most roads were in open prairie, where vulnerability to human detection was high. Accordingly, wolves avoided roads unless they were within high-canopy forests (Fig 2). In contrast, packs in the thickly-forested Whitefish Range selected roads (Fig 3), suggesting the benefit of easier travel in proximity to hiding cover outweighed the risk of encountering humans there. The density of roads in a pack's territory explained some of the variation in selection for roads, as packs in areas with high road densities not only used roads more (as would be expected even if habitat use were random), but also selected roads more strongly than packs with low road densities in their territory (Fig 3). A functional response in selection of this sort corroborates evidence that wolves may select roads to increase travel efficiency and facilitate predation, but avoid them in areas with low road densities where they may be perceived as unusual landscape features [87].

We hypothesized mountain lions would select hiding cover for stalking ungulate prey and avoiding humans, expecting them to select closed canopy vegetation, forested drainage bottoms, rugged areas, and to avoid roads. Mountain lions selected high canopy cover and avoided roads in the Garnet Range, supporting our hypothesis. Avoidance of roads may be attributed to risk of encountering humans, since human-caused mortalities like hunting and vehicle collisions are the leading source of mortality in many mountain lion populations [29,49]. Lack of cover for stalking prey [88], and lower densities of prey like elk near roads [89] may also help explain why mountain lions avoided roads. Importantly, selection of roads by mountain lions depends on environmental context, like surrounding topography and volume

of traffic [88]; we did not account for interactions between roads and other landscape features, and our results simply reflect the mean, study area-level habits of mountain lions. Associations between mountain lions and rugged terrain varied by study area, which decreased the generality of any single RSF. Importantly, different methods were used to relocate mountain lions in our study areas (GPS collars collecting data both day and night in the Garnet Range versus aerial VHF radiotelemetry conducted during daytime only in the other 2 areas), so differences in habitat use documented between study areas could be an artifice of telemetry methods and timing of data collection. Nonetheless, given the differences in prey communities between our study areas, we find it compelling that differences in use of rugged terrain by mountain lions was influenced by the types of prey they selected in each study area. On the Rocky Mountain Front, mountain lions used steep, rugged terrain (Fig 7B) associated with the habitat preferences of mule deer, elk, and bighorn sheep [66,90,91]. This result is intuitive, given that the diet biomass of mountain lions in this study area consisted primarily of elk (of either sex), bighorn sheep ewes, and mule deer bucks [66]. In contrast, mountain lions in the Garnet and Whitefish ranges selected moderately rugged terrain (Fig 7A), which is more characteristic of the habitat of white-tailed deer [92]. White-tailed deer were indeed the most killed prey for mountain lions in the Whitefish Range [59]. While mountain lion diet data were unavailable in the Garnet Range, ungulate harvest records indicate white-tailed deer were the most abundant prey in those areas (MFWP, unpublished data).

To be most effective, RSFs for predators should include direct measures of density of prey or habitat selection by prey [14]. However, the habitat prey select is dependent on predators, and vice versa, presenting a circular conundrum for researchers lacking predetermined information on either predator or prey density and distribution [93]. We lacked direct measurements of prey density or selection and were required to use proxies for prey encounter, presenting several major caveats to our findings. Firstly, attributing variation in mountain lion behavior to differences in prey communities is conjecture, as we did not use prey density or distribution data directly. This points to the limitations of proxies for prey encounter, as we cannot identify causative mechanisms underlying differences in predator space-use. Similarly, we cannot infer the effects of forage or prey density on habitat selection by wolves via vegetation cover-types alone. Cover-types were unreliable predictors of selection by wolves across ecosystem-types (Table 2), for several reasons. Cover-types are not perfect proxies for ungulate forage, as forage quality may vary at finer spatial scales than the level of a patch of a particular cover-type, so wolves could have selected points with high quality forage irrespective of cover-type. Secondly, even if a cover-type does contain high quality forage, it is not a perfect proxy for prey density, harkening to the ecological "space race" predators and their prey engage in [93]: in avoiding predators, prey may select cover-types with low amounts or quality of forage [80,94], so in tracking prey density, these cover-types may become irrelevant to predators [2,93,95]. By evidencing weak connections between vegetation cover-types and selection by wolves and mountain lions in different ecosystems, our findings serve as caution against using vegetation alone as proxies of predation risk for prey in ecological studies [96]. It is common practice to assume avoidance of open cover-types by ungulates is a predator avoidance strategy without actually measuring use of those cover-types by predators [23,97,98]; this could result in ecosystem-wide processes like trophic cascades or tradeoffs between forage and risk being falsely inferred.

Depending on local prey populations and where in the ecological space-race a predator-prey community is at, the features selected by a predator species may waver across its range. To expose general selection behaviors, it is important to assess resource selection by predators in multiple ecosystem-types. Our results suggest that features associated with the limitations of a predator's hunting mode may be more generalizable predictors of habitat selection than

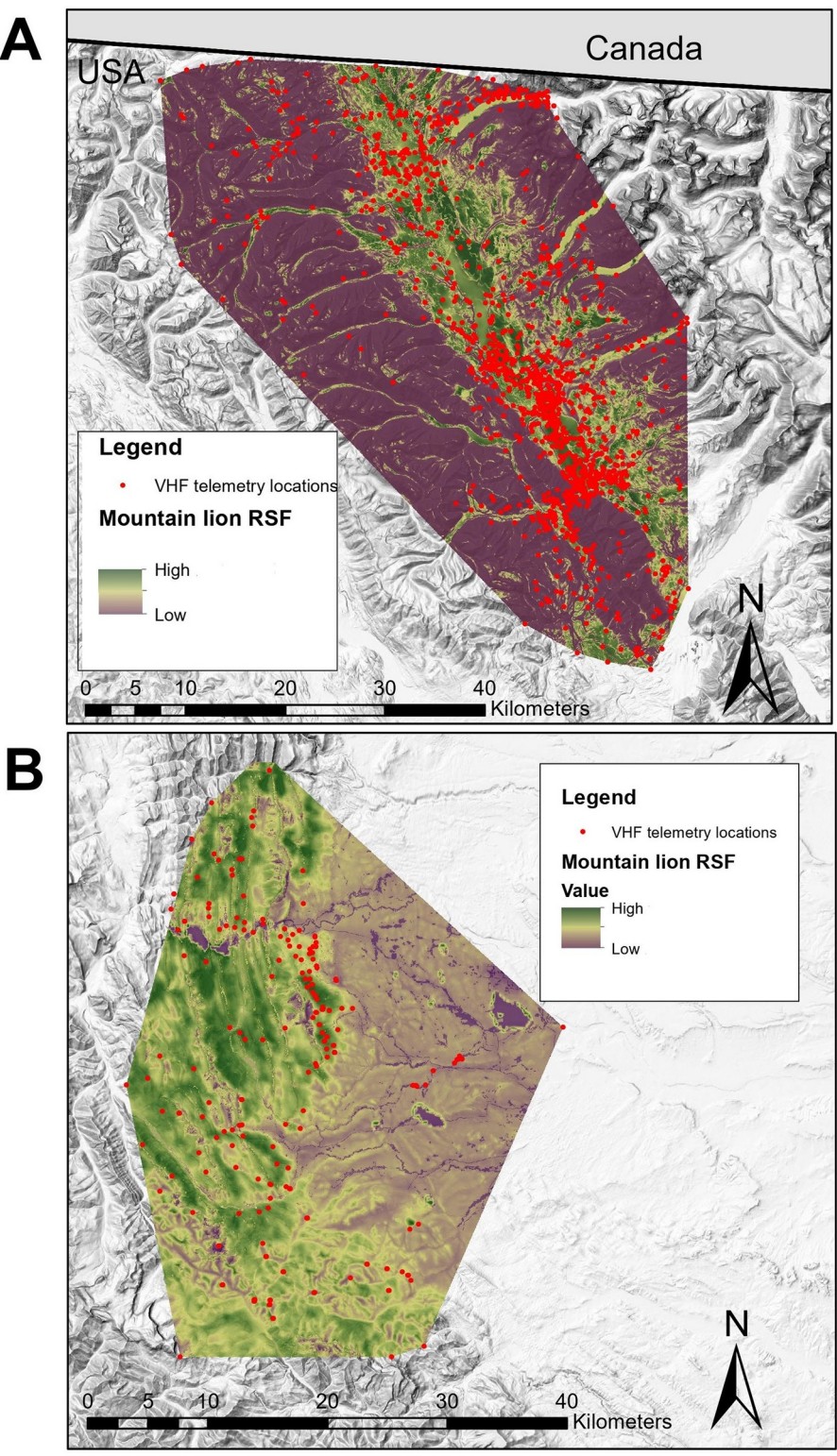

**Fig 7. Maps of mountain lion resource selection functions.** Predicted relative probability of selection from **(A)** 'moderate ruggedness' mountain lion resource selection function (RSF) in the Whitefish Range, MT, USA, and **(B)** 'high ruggedness' mountain lion RSF on the Rocky Mountain Front, MT, USA. RSF models were tested on VHF telemetry data from Kunkel et al. [59] and Williams [66]. Base map reprinted from Montana hillshade map under a CC BY license, with permission from Montana State Library, original copyright 2002.

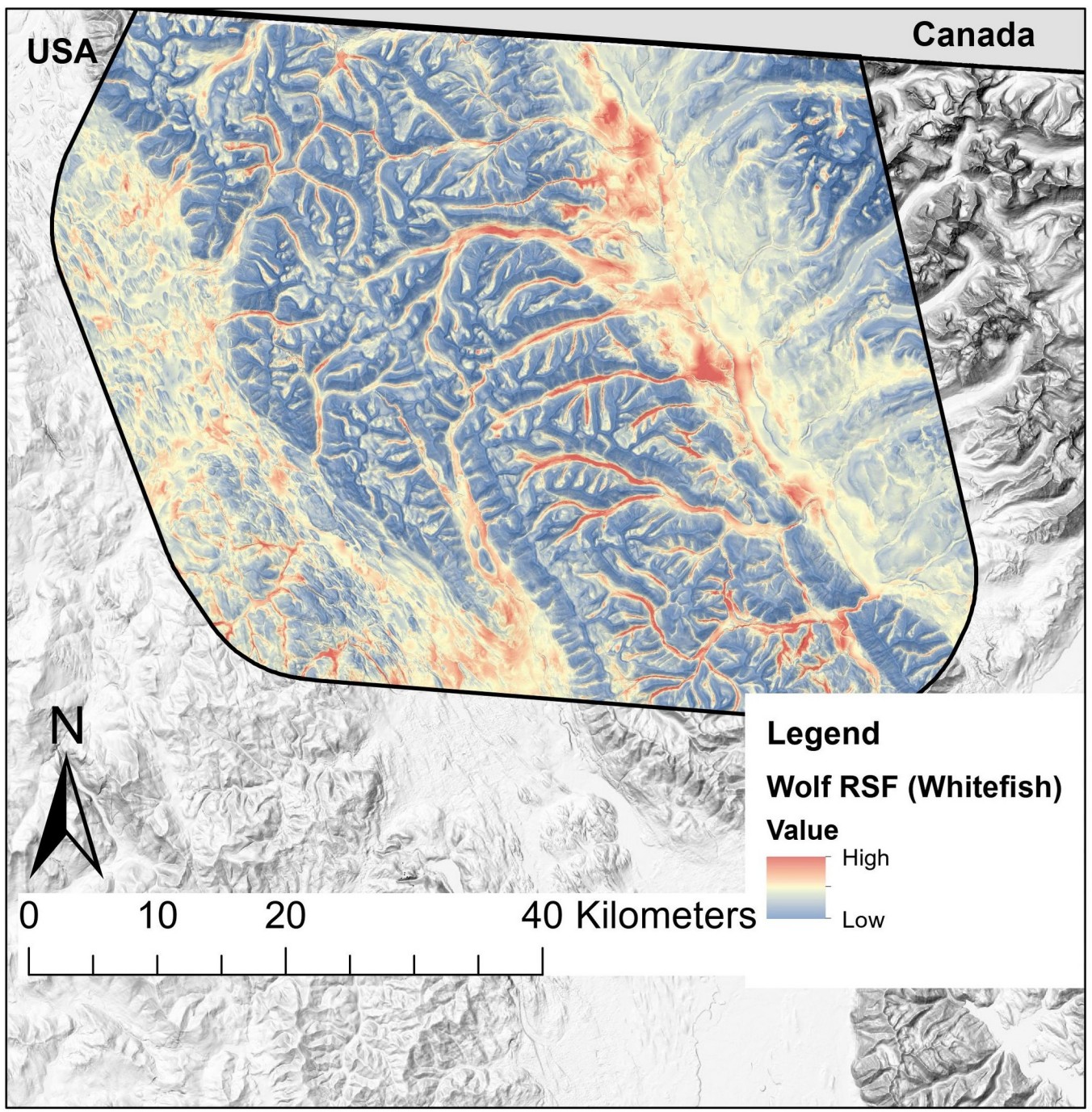

**Fig 8. Map of wolf resource selection function in the Whitefish Range.** Predicted relative probability of selection during summer from top-performing resource selection function (RSF) for wolves in the Whitefish Range, MT, USA. Reprinted from Montana hillshade map under a CC BY license, with permission from Montana State Library, original copyright 2002.

features indicative of local prey distributions. For example, wolves selected simple topography in all study areas, as their cursorial hunting mode is more effective in those areas [17]. As ambush predators, mountain lions are reliant on hiding cover for stalking, but are able to exploit prey in diverse topographic settings, like bighorn sheep in cliffy, rugged terrain [99],

guanacos (*Lama guanicoe*) in steppe [100], and beaver in riparian areas [101]. The plasticity of mountain lion behavior relative to topography was reflected in our analyses, as selection for rugged terrain varied between areas (Fig 7). Fine-scale features associated with hiding cover that is useful for stalking, like forest canopy, were more reliable predictors of habitat selection across variable prey settings than broad-scale topographic features.

There are numerous limitations to the inferences that can be drawn from our RSFs. Foremost, due to limitations in data availability, the timeframes in which mountain lion telemetry locations were collected (Rocky Mountain Front: 1991–1992; Whitefish Range: 1992–1996; Garnet Range: 2001–2006) did not overlap with the timeframe our vegetation cover-type data originated from (2017). Landcover composition changes rapidly in the American west [102], so the relationships we determined between mountain lions and vegetation cover-types may be inaccurate. Despite these discrepancies, our mountain lion RSFs are still useful, because physical features that change slowly with time (i.e. TPI, road density) were more important predictors of selection than cover-types (Table 3). In addition, we did not model mountain lion habitat selection directly in the Whitefish Range and Rocky Mountain Front, so our predictions cannot fully account for the true variability in mountain lion use of space in those areas. Changes in animal communities may also affect how well our RSFs apply in the present. For example, wolves colonized the Garnet Range in the midst of mountain lion monitoring in that study area [63], potentially inducing changes in habitat selection by mountain lions that our RSFs did not account for [103]. Finally, we did not account for potential spatial autocorrelation in our telemetry dataset, so we may have underestimated variances associated with the resource selection coefficients we report. Notwithstanding these caveats, our RSFs for wolves were generalizable to multiple ecosystem-types and should serve as useful indices of predation risk from wolves faced by ungulates in western Montana during summer. For example, wildlife managers are concerned with the population status of mule deer in areas like the Whitefish Range, and our RSFs for wolves could provide useful insight for understanding the risks faced by mule deer in that region (Fig 8). Our RSFs for mountain lions validated well in the Garnet and Whitefish Range, and performed fairly on the Rocky Mountain Front, and should serve as useful indicators of mountain lion predation risk in those regions. Before extrapolating our RSFs regions beyond our study system, we suggest testing their predictions against any available wolf or mountain lion location data first.

## Supporting information

**S1 File.**
(CSV)

**S2 File.**
(CSV)

## Acknowledgments

We are grateful to all those who spent countless hours in the field collecting data on wolves and mountain lions, and for agreeing to share the data used for these analyses. These include Diane Boyd, Wendy Cole, Jay Kolbe, Rich DeSimone, Bob Inman, Tyler Parks, Toni Ruth, Ty Smucker, and Jim Williams, among other important researchers and technicians. Hugh Robinson was instrumental in motivating this project early on, sharing data, and providing helpful reviews. Thanks also to Diane Boyd, Wendy Cole, Mark Hebblewhite, Joshua Millspaugh, Ty Smucker, David Walter, Michael Egan, and an anonymous reviewer for their helpful reviews.

## Author Contributions

**Conceptualization:** Collin J. Peterson, Michael S. Mitchell, Nicholas J. DeCesare, Chad J. Bishop, Sarah S. Sells.

**Data curation:** Collin J. Peterson, Sarah S. Sells.

**Formal analysis:** Collin J. Peterson, Nicholas J. DeCesare.

**Funding acquisition:** Nicholas J. DeCesare.

**Investigation:** Collin J. Peterson.

**Methodology:** Collin J. Peterson.

**Validation:** Collin J. Peterson.

**Visualization:** Collin J. Peterson.

**Writing – original draft:** Collin J. Peterson, Michael S. Mitchell, Nicholas J. DeCesare, Chad J. Bishop, Sarah S. Sells.

**Writing – review & editing:** Collin J. Peterson, Michael S. Mitchell, Nicholas J. DeCesare, Chad J. Bishop, Sarah S. Sells.

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
