## [Decision Letter · Decision Letter 0]

14 Apr 2021

PONE-D-21-03244

Habitat selection by wolves and mountain lions during summer in western Montana

PLOS ONE

Dear Dr. Peterson,

Thank you for submitting your manuscript to PLOS ONE. After careful consideration, we feel that it has merit but does not fully meet PLOS ONE’s publication criteria as it currently stands. Therefore, we invite you to submit a revised version of the manuscript that addresses the points raised during the review process.

We look forward to receiving your revised manuscript.

Kind regards,

W. David Walter, Ph.D.

Academic Editor

PLOS ONE

Journal Requirements:

In your Methods section, please provide additional location information of the study areas, including geographic coordinates for the data set if available.

We note that Figures 1, 5 and 6 in your submission contain map images which may be copyrighted. All PLOS content is published under the Creative Commons Attribution License (CC BY 4.0), which means that the manuscript, images, and Supporting Information files will be freely available online, and any third party is permitted to access, download, copy, distribute, and use these materials in any way, even commercially, with proper attribution. For these reasons, we cannot publish previously copyrighted maps or satellite images created using proprietary data, such as Google software (Google Maps, Street View, and Earth). For more information, see our copyright guidelines: http://journals.plos.org/plosone/s/licenses-and-copyright.

3a, You may seek permission from the original copyright holder of Figures 1, 5 and 6 to publish the content specifically under the CC BY 4.0 license. 

3b, If you are unable to obtain permission from the original copyright holder to publish these figures under the CC BY 4.0 license or if the copyright holder’s requirements are incompatible with the CC BY 4.0 license, please either i) remove the figure or ii) supply a replacement figure that complies with the CC BY 4.0 license. Please check copyright information on all replacement figures and update the figure caption with source information. If applicable, please specify in the figure caption text when a figure is similar but not identical to the original image and is therefore for illustrative purposes only.

Please include captions for *all* your Supporting Information files at the end of your manuscript, and update any in-text citations to match accordingly. Please see our Supporting Information guidelines for more information: http://journals.plos.org/plosone/s/supporting-information.

Additional Editor Comments:

I appreciate the author's efforts on this manuscript. Even though the reviews were positive, I believe the authors must address several additional concerns I have in their study design. I am not sure if the authors are unaware of this or dismiss it as insignificant so please address my concerns below in your resubmission.

1. The wolf dataset is from 2015-2108 on 18 individuals (some duplicates in a single pack, spread over 3 study areas) and the mountain lion dataset is from 2001-2006 on only 17 individuals. Do the authors believe that comparison of datasets over 10 years apart is irrelevant and not worth addressing the issues that arise from these datasets representing different periods?

2. The authors appear to use the same vegetation layer for wolf data (2015-2018) and mountain lion data (2001-2006) from a state landcover map that is available only starting in 2010 according the citation (60) referenced by the authors (http://geoinfo.msl.mt.gov/msdi/land_use_land_cover). The authors should justify or at least discuss the issues of having a dataset from 2010 representing landcover from 10 years prior to when the landcover layer was created. The methods here must be more detailed and include some form of validation to use of this vegetation layer dataset for both wolf and mountain lion RSFs. Table 1 is not enough as readers should not have to investigate data sources not described by the authors in their text.

3. If the authors are citing previous manuscripts that used the mountain lion datasets, what novel findings are the authors conducting with this data? Please address this in some manner in your Methods for these mountain lion datasets after citing them.

Reviewers' comments:

Reviewer's Responses to Questions

**Comments to the Author**

1. Is the manuscript technically sound, and do the data support the conclusions?

Reviewer #1: Yes

Reviewer #2: Yes

2. Has the statistical analysis been performed appropriately and rigorously? 

Reviewer #1: Yes

Reviewer #2: Yes

3. Have the authors made all data underlying the findings in their manuscript fully available?

Reviewer #1: Yes

Reviewer #2: Yes

4. Is the manuscript presented in an intelligible fashion and written in standard English?

Reviewer #1: Yes

Reviewer #2: Yes

5. Review Comments to the Author

Reviewer #1: Overall, I found this an interesting and easy-to-read manuscript. I appreciate the multi-species multi-study area aspect of the study – these types of studies are very useful for understanding generalizations in ecological processes but aren’t often conducted (usually because collecting these types of data is difficult!). Although I think this manuscript has potential, there are several things that I think need to be improved on. I have a few general comments about the framing of the manuscript (especially the discussion) as well as some specific comments in the text.

General Comments:

I think that the introduction does a nice job at setting up the general relevance of the study – e.g. various environmental features are often used as proxies of predator distribution, but not much research has focused on how generalizable these proxies are. This general framing of the paper gets lost by the discussion. The discussion is very species-focused (e.g. what habitats did wolves select for and why, what habitats did mountain lions select for and why etc.). Although there are some sections that attempt to link the results back to the broader idea (e.g. line 433 suggests caution in using vegetation as a proxy for wolf distribution), I think the paper could be strengthened by adding a few concluding paragraphs at the end of the discussion. Specifically, these paragraph(s) should broaden out from being wolf/mountain lion focused and instead the relevance of these results outside of this study: What does this research tell us (generally) about using proxies for predator distribution? What different things should researchers consider if they are thinking of using a proxy for predators in their research? Why is understanding the generalization of proxies important? Etc. Similarly, I think the abstract could use a sentence or two about the broader idea of using proxies (e.g. it’s currently too wolf/mountain lion focused).

There is currently a strong focus on how wolf/mountain lion habitat selection might be driven by characteristics related to prey availability or hunting behaviors, with a slightly lesser focus on how humans might be driving habitat selection patterns. While it is true that prey/hunting might be a driver of predator habitat selection, there are other ecological factors that might affect habitat use (e.g. suitable habitat for den sites, territorial boundaries, water availability etc.). Although wolf locations were removed to try to eliminate “non traveling” locations, this same procedure was not replicated for the mountain lion data. Therefore, care needs to be taken when discussing the mountain lion results especially, since the used locations contain both traveling and non-traveling locations. I don’t think you need to re-run any analyses, but the discussion should ideally bring up the fact that the mountain lion results are not necessarily indicative of selection for hunting habitats/prey availability alone and instead might also indicate selection for non-hunting related features.

One component of the study design that isn’t really discussed in potential biases of GPS vs. VHF data for the mountain lions. My main concern is that lack of generalization across study areas might be an artifact of data collection. For example, I’m assuming that the VHF locations were only collected during the day, whereas the GPS locations were collected day and night. If there are any differences in mountain lion selection between night and day, using the VHF data as validation data would likely cause poor validation results. Similarly, given that VHF data usually requires humans to collect the data, there can sometimes be spatial biases if there are parts of the study area inaccessible to humans, in which case the “used” VHF data might not be representative of what the animals are actually using. I recognize that you used the best data you have available, and that suitable GPS data across several study areas is often hard to come by. You mention in the discussion that selection for different TRI in different areas could be related to differences in prey availability, which is definitely a possibility, but I think it could also be valuable to mention that the poor validation could be an artifact of differences in data collection techniques.

Specific Comments:

Table 1. In the introduction (Lines 103-104), you talk about how wolves might select for roads because they might be energetically efficient for hunting. In Table 1, wolves are predicted to select for roads in order to “increase prey encounters.” Increasing prey encounters and being energetically advantageous for movement seem like two different explanations, although they both seem valid. I would add a few sentences in the introduction about why roads would increase prey encounters – are prey more common on roads as well?

Line 227: The road covariate is described as “road density.” Given that you presumably extracted covariates from point locations, what was the “area” used to calculate this density, since calculating point-level density seems difficult? For example, if the road density was calculated for each study area (like is described on lines 232-233) that would mean that all locations within the study area would be associated with the same density and therefore there wouldn’t be any variation in selection to model. Also, a better explanation of the road data itself is needed – did you use all roads, just paved roads, just unpaved roads etc.?

Line 254: I’m confused by the decision to run mixed effects models to only look at the effects of roads, whereas the vegetation/habitat features were modeled separately by wolf pack. Why not run mixed effect models for all covariates of interest? There needs to be better justification for why the analyses included fixed effect models for each pack for some covariates vs. mixed effect models for road effects.

Lines 287 – 288: Did you visually look at where locations occurred or actually extract the ruggedness covariate from the VHF locations/statistically test that used locations differed in ruggedness between the two areas (e.g. what do you mean by “we observed” [Line 288])? If it was in fact visually looking at where the locations were, I would recommend generating actual statistics to back up this statement – humans are prone to observing patterns in data that actually aren’t meaningful.

Line 297: I’ve been working on some cross-validation work recently and was interested in the R package that you used (kxvglm) for your internal cross-validation since I had never heard of it. A google search yielded no results for this package and the citation in your manuscript (Boyce 2002) also made no mention of this package. To make sure that interested readers are able to follow your methods and find this package, can you update the citation or package information in your manuscript? Maybe the package has changed names?

Line 450: Since one of hypotheses of this manuscript is that wolf/lion habitat use is affected by probability of encountering humans (i.e. using roads as a proxy for humans), I would add a few sentences discussing the observed effect of roads on mountain lions. The mountain lion section of the discussion is currently very prey/hunting focused.

Figures 4 and 5: I think these figures are fine, but I think the 2 supplemental figures actually add more to the results section than these two figures.

Reviewer #2: Major Comments:

In general, I believe the methods have been performed appropriately and the conclusions are sound. The authors took appropriate measures to clean and prepare data for analysis in accordance with what is necessary for an RSF. Additionally, the approach used to build and validate models was thorough and provides sufficient confidence in their results. Most importantly the authors limit their analysis to datasets with sufficient samples for their analysis, specifically withholding mountain lion datasets at sites with a small number of locations. One thing that may need to be noted in their methods is if the authors tested for autocorrelation in their data.

One potential issue that I believe should be noted in the conclusions relates to the choice of variables. As noted, successful predation can be broken into the chance of encounter and chance of successful predatory encounter. Variables related to both of these factors were chosen for wolves, but variables related to chance of encounter for mountain lions do not seem to have been given equal focus. In table 1, there are several variables predicted to be neutral with respect to habitat selection. In general, differences in the data available and variables could affect the interpretation of these results. Additional discussion of the implications of limitations of location data for mountain lions and the proxies used for models could be added to the discussion.

Line Comments:

57-60: This entire paragraph seems focused on predation risk from the perspective of the prey which contrasts somewhat with the predator focus of the paper, particularly this sentence. Readers may be able to follow the logic behind the proxies that were selected if this sentence focused less on the factors that present risk to prey and more on the factors leading to successful prey capture by predators.

78-79: How does probability of encountering prey factor in?

Table 1: I think the terms hypothesis and prediction could be switched in this table. Your prediction should be the direction of the relationship you expect and your hypothesis should be the reason for that prediction.

197-207: There is a longer time between mountain lion captures at the 3 study sites than between wolf captures. Are there any potential differences in the study areas between time periods (land cover changes, prey populations) that could have impacted results between periods and did you account for those potential differences?

220-233: Since these variables were chosen because they may be proxies for prey encounter and capture, did you explicitly consider habitat associations of prey in the area? If data these specifically is poor in this study areas these choices are understandable, but I wonder about some other variables that could impact the probability of use by ungulates (for example NDVI).

234: Did you test for spatial autocorrelation in variables before fitting RSFs? Tests for autocorrelation or reasons why you don’t think it is present in this dataset may be appropriate here.

247-248: should this sentence read “If removal of a main effect decreased AIC …”. If removing a main effect increases AIC, then the model with that variable has better fit and that main effect should be included regardless of the interaction.

281-282: Why use the same variables for mountain lion as wolves, particularly the ones that you thought would be unrelated to mountain lion selection?

316: You have no table 2.

428-436: The vegetation classes used here are only proxies for ungulate use and you don’t really have any data on ungulates. This conclusion with relation to ungulates seems too strong given that we aren’t really sure of the quality of the forage or to what extent ungulates are foraging in that cover type in this area. I think this paragraph in particular should be revised.

450-467: I think the suggestion of differences in prey between the study sites is reasonable, but specific implications about different prey habitat selection is founded on a few assumptions about what habitat types prey are using in these areas. This may be a good place to discuss the potential limitations of using proxies for prey encounter.

6. PLOS authors have the option to publish the peer review history of their article (what does this mean?). If published, this will include your full peer review and any attached files.

Reviewer #1: No

Reviewer #2: **Yes: **Michael Egan

---

## [Author Response · Author response to Decision Letter 0]

24 May 2021

Reviewers' comments:

Reviewer's Responses to Questions

Comments to the Author

1. Is the manuscript technically sound, and do the data support the conclusions?

Reviewer #1: Yes

Reviewer #2: Yes

2. Has the statistical analysis been performed appropriately and rigorously?

Reviewer #1: Yes

Reviewer #2: Yes

3. Have the authors made all data underlying the findings in their manuscript fully available?

Reviewer #1: Yes

Reviewer #2: Yes

4. Is the manuscript presented in an intelligible fashion and written in standard English?

Reviewer #1: Yes

Reviewer #2: Yes

5. Review Comments to the Author

Reviewer #1: Overall, I found this an interesting and easy-to-read manuscript. I appreciate the multi-species multi-study area aspect of the study – these types of studies are very useful for understanding generalizations in ecological processes but aren’t often conducted (usually because collecting these types of data is difficult!). Although I think this manuscript has potential, there are several things that I think need to be improved on. I have a few general comments about the framing of the manuscript (especially the discussion) as well as some specific comments in the text.

General Comments:

I think that the introduction does a nice job at setting up the general relevance of the study – e.g. various environmental features are often used as proxies of predator distribution, but not much research has focused on how generalizable these proxies are. This general framing of the paper gets lost by the discussion. The discussion is very species-focused (e.g. what habitats did wolves select for and why, what habitats did mountain lions select for and why etc.). Although there are some sections that attempt to link the results back to the broader idea (e.g. line 433 suggests caution in using vegetation as a proxy for wolf distribution), I think the paper could be strengthened by adding a few concluding paragraphs at the end of the discussion. Specifically, these paragraph(s) should broaden out from being wolf/mountain lion focused and instead the relevance of these results outside of this study: What does this research tell us (generally) about using proxies for predator distribution? What different things should researchers consider if they are thinking of using a proxy for predators in their research? Why is understanding the generalization of proxies important? Etc. Similarly, I think the abstract could use a sentence or two about the broader idea of using proxies (e.g. it’s currently too wolf/mountain lion focused).

There is currently a strong focus on how wolf/mountain lion habitat selection might be driven by characteristics related to prey availability or hunting behaviors, with a slightly lesser focus on how humans might be driving habitat selection patterns. While it is true that prey/hunting might be a driver of predator habitat selection, there are other ecological factors that might affect habitat use (e.g. suitable habitat for den sites, territorial boundaries, water availability etc.). Although wolf locations were removed to try to eliminate “non traveling” locations, this same procedure was not replicated for the mountain lion data. Therefore, care needs to be taken when discussing the mountain lion results especially, since the used locations contain both traveling and non-traveling locations. I don’t think you need to re-run any analyses, but the discussion should ideally bring up the fact that the mountain lion results are not necessarily indicative of selection for hunting habitats/prey availability alone and instead might also indicate selection for non-hunting related features.

CJP: Our decision not to eliminate “non-traveling” locations of mountain lions for RSF analyses was informed by evidence that fine-scale habitat selection by mountain lions does not vary substantially when mobile versus stationary. For example, Blake and Gese (2016) found little differences in the habitat characteristics of mountain lion kill-sites and moving locations in southern Montana, and concluded that mountain lions are generally “in hunting mode” while moving through their location. Further, Smith et al. (2019) found that habitat selection by mountain lions does not vary with time of day. Filtering out non-moving locations is less important for mountain lions than for wolves, since mountain lions do not aggregate at rendezvous sites during the summer as wolves do. Regarding the filtering of wolf locations, we added to our methods:

“This is an important procedure to for focusing on hunting behavior of wolves, because wolves spend much of their time at rendezvous sites during the summer…”, 

and for lions, we added: 

“We developed 3rd order fixed-effect RSFs for mountain lions in the Garnet Range, and included all locations because 3rd order mountain lion habitat selection does not vary substantially when mobile versus stationary (20,73), and mountain lions do not concentrate activity at rendezvous sites like wolves do.”

Given the evidence that mountain lions are generally “in hunting mode”, we felt that human encounter risk was the next most important factor driving space-use. By assessing selection for roads, we did address how non-hunting related factors influence space-use, but we will certainly provide more discussion regarding mountain lions relations towards roads. 

One component of the study design that isn’t really discussed in potential biases of GPS vs. VHF data for the mountain lions. My main concern is that lack of generalization across study areas might be an artifact of data collection. For example, I’m assuming that the VHF locations were only collected during the day, whereas the GPS locations were collected day and night. If there are any differences in mountain lion selection between night and day, using the VHF data as validation data would likely cause poor validation results. Similarly, given that VHF data usually requires humans to collect the data, there can sometimes be spatial biases if there are parts of the study area inaccessible to humans, in which case the “used” VHF data might not be representative of what the animals are actually using. I recognize that you used the best data you have available, and that suitable GPS data across several study areas is often hard to come by. You mention in the discussion that selection for different TRI in different areas could be related to differences in prey availability, which is definitely a possibility, but I think it could also be valuable to mention that the poor validation could be an artifact of differences in data collection techniques.

CJP: Luckily, the majority of VHF re-locations were collected from the air, which reduces some of the spatial biases associated with relocating animals. However, Williams (1992) did investigate the spatial error of his VHF relocations and estimated them to be accurate within 150-200m, which is substantial, especially relative to fine-scale habitat features, so differences in habitat use between study areas could be an artifice of collection methods. In the discussion, we added:

 “Importantly, different methods were used to relocate mountain lions in our study areas (GPS collars collecting data both day and night in the Garnet Range versus aerial VHF radiotelemetry conducted during daytime only in the other 2 areas), so differences in habitat use documented between study areas could be an artifice of telemetry methods and timing of data collection. Nonetheless, given the differences in prey communities between our study areas, we find it compelling that differences in use of rugged terrain by mountain lions was influenced by the types of prey selected in each study area.”

Specific Comments:

Table 1. In the introduction (Lines 103-104), you talk about how wolves might select for roads because they might be energetically efficient for hunting. In Table 1, wolves are predicted to select for roads in order to “increase prey encounters.” Increasing prey encounters and being energetically advantageous for movement seem like two different explanations, although they both seem valid. I would add a few sentences in the introduction about why roads would increase prey encounters – are prey more common on roads as well?

CJP: Good point. By traveling on roads, wolves can travel faster and farther, allowing them to increase their encounter rate with prey. To the introduction, we added:

“Roads can increase risk of encountering hunters, trappers, and vehicles, but may also serve as energetically efficient travel routes while hunting, allowing wolves to travel farther and faster, increasing their encounter rate with prey (41,42).” 

We also noted that roads make for energetically efficient travel in Table 1.

Line 227: The road covariate is described as “road density.” Given that you presumably extracted covariates from point locations, what was the “area” used to calculate this density, since calculating point-level density seems difficult? For example, if the road density was calculated for each study area (like is described on lines 232-233) that would mean that all locations within the study area would be associated with the same density and therefore there wouldn’t be any variation in selection to model. Also, a better explanation of the road data itself is needed – did you use all roads, just paved roads, just unpaved roads etc.?

CJP: We noted in the methods that our metric of road density was “…total length of roads (km) per km2”, and specified that this included both paved and unpaved roads. These road data have a 1km2 resolution.

Line 254: I’m confused by the decision to run mixed effects models to only look at the effects of roads, whereas the vegetation/habitat features were modeled separately by wolf pack. Why not run mixed effect models for all covariates of interest? There needs to be better justification for why the analyses included fixed effect models for each pack for some covariates vs. mixed effect models for road effects.

CJP: In modeling wolf resource selection, we were balancing 2 objectives: predicting wolf behavior across diverse ecosystems and understanding the drivers of variation in wolf behavior. It is difficult to form spatially-explicit predictions with mixed-effects RSFs. To simplify our RSF predictions, we prioritized study area-specific fixed-effects models and only used mixed-effects models for understanding the context-specific influence of roads. We felt that roads were the most important variable to do this with and warranted deeper investigation because they served as our sole indicator of human encounter risk, and there is ample evidence that the relationship of wolves to roads is typically dependent on environmental context. 

We justified our exploration of the functional response to roads further in the introduction, saying: 

“Wolves may respond differently to roads at different road densities (i.e., a functional response in selection (44–46)), reflecting a context-dependent tradeoff between the risk of encountering humans and the benefits of efficient travel and access to prey (47). Thus, it is particularly important to investigate functional responses in selection for roads by wolves to understand how they attenuate exposure to risk (45).”

In our methods, we added:

“We were balancing multiple objectives of predicting wolf behavior across multiple ecosystems and understanding the drivers of variation in wolf behavior. Thus, we focused the bulk of our analyses on developing study area-specific fixed-effects models, which are simple for making spatially-explicit predictions of resource selection.”

We agree, including random slopes for other variables of interest (e.g. vegetation cover-types, canopy cover, etc.) would have been helpful for explaining variation in habitat selection between packs, which we acknowledge in the discussion, saying: 

“For wolves, selection of vegetation cover-types associated with higher forage quality for ungulates was inconsistent across regions, suggesting that for ungulates, the risk of encountering wolves may be decoupled from forage dynamics. This may relieve elk and deer from having to make tradeoffs between forage and security from wolves during summer. Selection for vegetation cover-types by wolves may depend on the density of certain cover-types (e.g. timber harvests) in a region (45). A functional response in selection of this nature could underlie the variance in selection for cover-types by wolves we observed, which we did not test for. However, given the consistent, strong effects of slope and TPI across study areas, our results suggest that broad-scale avoidance of drainage bottoms and low slopes by ungulates may be more effective than avoiding open vegetation cover-types to circumvent predation risk from wolves in western Montana.”

Lines 287 – 288: Did you visually look at where locations occurred or actually extract the ruggedness covariate from the VHF locations/statistically test that used locations differed in ruggedness between the two areas (e.g. what do you mean by “we observed” [Line 288])? If it was in fact visually looking at where the locations were, I would recommend generating actual statistics to back up this statement – humans are prone to observing patterns in data that actually aren’t meaningful.

CJP: We initially observed this pattern visually, which prompted us to investigate differences in TRI quantitatively. We included in our results:

 “Mean TRI at locations used by mountain lions on the Rocky Mountain Front (x ® = 80.882, SD = 30.299) was 1.76 times higher than in the Whitefish Range (x ® = 45.85, SD = 43.266) and 1.22 times higher than in the Garnet Range (x ® = 66.213, SD = 27.244).” 

In our results, we added:

 “We initially observed that mountain lions in the Whitefish Range used less rugged terrain than on the Rocky Mountain Front visually, and confirmed this by comparing mean TRI at used locations of mountain lions in all study areas.”

Line 297: I’ve been working on some cross-validation work recently and was interested in the R package that you used (kxvglm) for your internal cross-validation since I had never heard of it. A google search yielded no results for this package and the citation in your manuscript (Boyce 2002) also made no mention of this package. To make sure that interested readers are able to follow your methods and find this package, can you update the citation or package information in your manuscript? Maybe the package has changed names?

CJP: It was an error on my part to refer to kxvglm as a package. Kxvglm is simply a function written in R that was shared with me by someone in the Boyce lab, via a professor of mine in graduate school. I cited Boyce (2002) for our approach towards conducting k-folds cross validation because that study justified the validity of that method. I would be happy to share the code we used to conduct k-folds cross validation with my submission upon request. The code I received specifically states I am able to redistribute or modify it under the terms of the GNU General Public License as published by the Free Software Foundation.

Line 450: Since one of hypotheses of this manuscript is that wolf/lion habitat use is affected by probability of encountering humans (i.e. using roads as a proxy for humans), I would add a few sentences discussing the observed effect of roads on mountain lions. The mountain lion section of the discussion is currently very prey/hunting focused.

CJP: In the discussion, I added a couple sentences addressing the hypothesized mechanisms driving avoidance of roads: lack of hiding cover for stalking prey, avoidance of roads by important prey like elk, and risk of vehicle collisions.

Figures 4 and 5: I think these figures are fine, but I think the 2 supplemental figures actually add more to the results section than these two figures.

CJP: I added supplementary figures 1 and 2 to the main manuscript. They are now figures 3 and 5, respectively.

Reviewer #2: Major Comments:

In general, I believe the methods have been performed appropriately and the conclusions are sound. The authors took appropriate measures to clean and prepare data for analysis in accordance with what is necessary for an RSF. Additionally, the approach used to build and validate models was thorough and provides sufficient confidence in their results. Most importantly the authors limit their analysis to datasets with sufficient samples for their analysis, specifically withholding mountain lion datasets at sites with a small number of locations. One thing that may need to be noted in their methods is if the authors tested for autocorrelation in their data.

One potential issue that I believe should be noted in the conclusions relates to the choice of variables. As noted, successful predation can be broken into the chance of encounter and chance of successful predatory encounter. Variables related to both of these factors were chosen for wolves, but variables related to chance of encounter for mountain lions do not seem to have been given equal focus. In table 1, there are several variables predicted to be neutral with respect to habitat selection. In general, differences in the data available and variables could affect the interpretation of these results. Additional discussion of the implications of limitations of location data for mountain lions and the proxies used for models could be added to the discussion.

CJP: I am glad you noticed that the relationships we hypothesized between landscape variables and habitat selection by mountain lions focused less on chance of encounter with prey than our hypotheses for wolves. These hypotheses were spawned from the notion that variables associated with the probability of prey encounter probability may have a stronger effect on wolf habitat selection than mountain lion selection, due to their different hunting modes. In the introduction, we added:

“For every pursuit wolves engage in, they have a relatively low probability of capturing prey (39), thus, selecting habitat that maximizes encounters with prey increases the hunting success of wolves (38)… (Mountain lions) often select structurally complex, rugged topography that provides fine-scale hiding cover like boulders and outcrops (48), but will also select dense vegetation cover-types like thick forests and riparian areas within topographically simple areas (18–20,49). Thus, mountain lions may exhibit weaker selection for features associated with prey encounter than wolves (17,38).”

Line Comments:

57-60: This entire paragraph seems focused on predation risk from the perspective of the prey which contrasts somewhat with the predator focus of the paper, particularly this sentence. Readers may be able to follow the logic behind the proxies that were selected if this sentence focused less on the factors that present risk to prey and more on the factors leading to successful prey capture by predators.

CJP: I disagree with the statement that this paragraph seems focused on predation risk from the perspective of prey. In the second sentence, we state that predation risk is the consequence of habitat selection by predators. We end the paragraph by foreshadowing our goal of “understanding how predators select habitat…” Our goal of this paragraph was not to justify specific methods and variables we tested. Rather, we intended to highlight the value in understanding how predators select habitat because of the numerous ecological consequences the imposition of risk on prey through use of space by predators can have. I feel paragraph #3 in the introduction justifies use of the proxies we chose sufficiently.

78-79: How does probability of encountering prey factor in?

CJP: Added, “…open, less-rugged terrain may signify areas where cursorial predators like wolves (Canis lupus) can maximize opportunities to detect prey (15,16), and topographical features like drainages may enhance prey capture (17).”

Table 1: I think the terms hypothesis and prediction could be switched in this table. Your prediction should be the direction of the relationship you expect and your hypothesis should be the reason for that prediction.

CJP: Done.

197-207: There is a longer time between mountain lion captures at the 3 study sites than between wolf captures. Are there any potential differences in the study areas between time periods (land cover changes, prey populations) that could have impacted results between periods and did you account for those potential differences?

CJP: We did not account for these differences in our analyses, and it is one of the major caveats of this research. We address the issues that may arise from mismatched telemetry and landcover data at the end of our discussion. We also address how comparisons of telemetry data from different time periods can be problematic due to changes in ecological communities (e.g. wolves colonizing an area and potentially affecting mountain lion habitat selection).

220-233: Since these variables were chosen because they may be proxies for prey encounter and capture, did you explicitly consider habitat associations of prey in the area? If data these specifically is poor in this study areas these choices are understandable, but I wonder about some other variables that could impact the probability of use by ungulates (for example NDVI).

CJP: We hypothesized that open habitat (grasslands, shrublands, low canopy cover) would contain higher quality forage for ungulates, therefore would increase prey encounter rates. We cite several studies connecting deer and elk habitat selection and forage quality to the landcover variables we were interested in (Ager et al. 2003; Proffitt et al. 2016). I agree, NDVI would have been a useful proxy for prey encounter, however, it was difficult to obtain and process NDVI data and we resorted to landcover variables for simplicity. We exhibited restraint in interpreting lack of selection for open vegetation cover-types as lack of selection for areas where wolves or mountain lions are likely to encounter prey. We took lengths to address this in the first and fifth paragraphs of our discussion. 

234: Did you test for spatial autocorrelation in variables before fitting RSFs? Tests for autocorrelation or reasons why you don’t think it is present in this dataset may be appropriate here.

CJP: We did not test for spatial autocorrelation. By removing non-traveling locations of wolves from our dataset, we decreased clustering of wolf GPS locations in space, reducing spatial autocorrelation. Unfortunately, we did not account for spatial autocorrelation in our mountain lion telemetry dataset. Within our caveats in the discussion, we acknowledge that we may have underestimated the variances associated with resource selection coefficients by failing to account for spatial autocorrelation.

247-248: should this sentence read “If removal of a main effect decreased AIC …”. If removing a main effect increases AIC, then the model with that variable has better fit and that main effect should be included regardless of the interaction.

CJP: You are correct, I changed it.

281-282: Why use the same variables for mountain lion as wolves, particularly the ones that you thought would be unrelated to mountain lion selection?

CJP: For both wolves and mountain lions, we hypothesized that these variables would be relevant to prey species and would correlate with either the probability of encountering or killing prey. Though we predicted several of these variables to have no effect on mountain lion selection, we found it necessary to test those variables to determine whether they were useful proxies for risk, and to better understand how mountain lions select habitat relative to attributes relevant to their prey. Even a result of no effect is interesting in and of itself; for example, that mountain lions did not select cover-types hypothesized to contain high quality forage is informative and helps us understand the ultimate drivers of mountain lion behavior.

316: You have no table 2.

CJP: Oops! I mis-labeled Table 2 as Table 3.

428-436: The vegetation classes used here are only proxies for ungulate use and you don’t really have any data on ungulates. This conclusion with relation to ungulates seems too strong given that we aren’t really sure of the quality of the forage or to what extent ungulates are foraging in that cover type in this area. I think this paragraph in particular should be revised.

CJP: We addressed the caveats that come with using landcover data as proxies for either forage or prey density, in detail:

“Cover-types were unreliable predictors of selection by wolves across ecosystem-types (Table 2), for several reasons. Cover-types are not perfect proxies for ungulate forage, as forage quality may vary at finer spatial scales than the level of a patch of a particular cover-type, so wolves could have selected points with high quality forage irrespective of cover-type. Secondly, even if a cover-type does contain high quality forage, it is not a perfect proxy for prey density, harkening to the ecological “space race” predators and their prey engage in [2]. In avoiding predators, prey may select cover-types with low amounts or quality of forage [79,93], so in tracking prey density, these cover-types may become irrelevant to predators [2,92,94]. By evidencing weak connections between vegetation cover-types and selection by wolves and mountain lions in different ecosystems, our findings serve as caution against using vegetation alone as proxies of predation risk for prey in ecological studies [95]. It is common practice to assume avoidance of open cover-types by ungulates is a predator avoidance strategy without actually measuring use of those cover-types by predators [23,96,97]; this could result in ecosystem-wide processes like trophic cascades or tradeoffs between forage and risk being falsely inferred.

450-467: I think the suggestion of differences in prey between the study sites is reasonable, but specific implications about different prey habitat selection is founded on a few assumptions about what habitat types prey are using in these areas. This may be a good place to discuss the potential limitations of using proxies for prey encounter.

Agreed, we modified our discussion substantially, addressing how proxies are imperfect indicators of prey due to the ecological space-race predators and prey are in. In the discussion, we added:

CJP: “To be most effective, RSFs for predators should include direct measures of density or habitat use or selection of prey [14]. However, the habitat prey select is dependent on predators, and vice versa, presenting a circular conundrum for researchers lacking predetermined information on either predator or prey density and distribution [92]. We lacked direct measurements of prey density or selection and were required to use proxies for prey encounter, presenting several major caveats. Firstly, attributing variation in mountain lion behavior to differences in prey communities is conjecture, as we did not use prey density or distribution data directly. This points to the limitations of proxies for prey encounter, as we cannot identify causative mechanisms underlying differences in predator space-use. Similarly, we cannot infer the effects of forage or prey density on habitat selection by wolves via vegetation cover-types alone…”

---

## [Editor Report · Decision Letter 1]

15 Jun 2021

PONE-D-21-03244R1

Habitat selection by wolves and mountain lions during summer in western Montana

PLOS ONE

Dear Dr. Peterson,

Thank you for submitting your manuscript to PLOS ONE. After careful consideration, we feel that it has merit but does not fully meet PLOS ONE’s publication criteria as it currently stands. Therefore, we invite you to submit a revised version of the manuscript that addresses the points raised during the review process.

We look forward to receiving your revised manuscript.

Kind regards,

W. David Walter, Ph.D.

Academic Editor

PLOS ONE

Journal Requirements:

Additional Editor Comments (if provided):

I appreciate the authors response to reviewers, specifically on the caveats of using GPS and VHF datasets in your analyses. As I read through both drafts of your manuscript, I shared the concerns of Reviewer #1 that the authors chose to assess resource selection of 2 predators collected with 2 technologies (GPS and VHF) using a single method (3rd order RSF) across numerous study areas. While those methods would be appropriate for the limited dataset collected with VHF technology, there are more appropriate methods for use with GPS technology. Justification for the study design was provided in Lines 227-230 that "...none of these studies assessed resource selection via a used-available design..." It seems that another study could be done in the future using only your GPS collar data from both species and the justification would be similar to the authors by stating "......none of these studies assessed resource selection via a step selection approach..." or similar reasoning. Is there a particular reason the authors chose RSF over more detailed methods using step selection or individual location-specific approaches with only the GPS datasets? Do the authors feel their study design and methods on several study areas is more reliable and beneficial than on a study design using only GPS-collared predators?

Line 154: remove plural for "lion" but add plural for "area".

Line 158: Change "The Garnets" to the "Garnett Range" which is should be referred to as throughout for consistency (also Line 163).

Line 160: Remove "Garnet" from mountain lions considering this is mentioned in the paragraph describing the Garnett Range.

Lines 223-226: The author identify the "MFWP's biomedical protocol for free-ranging mountain lions" and wolves but only report University IACUC #A3327-01 for mountain lions in the Whitefish Range. Were any other Institutional Animal Care and Use Protocols conducted for lions or wolves for any of the other studies?

---

## [Author Response · Author response to Decision Letter 1]

24 Jun 2021

I appreciate the authors response to reviewers, specifically on the caveats of using GPS and VHF datasets in your analyses. As I read through both drafts of your manuscript, I shared the concerns of Reviewer #1 that the authors chose to assess resource selection of 2 predators collected with 2 technologies (GPS and VHF) using a single method (3rd order RSF) across numerous study areas. While those methods would be appropriate for the limited dataset collected with VHF technology, there are more appropriate methods for use with GPS technology. Justification for the study design was provided in Lines 227-230 that "...none of these studies assessed resource selection via a used-available design..." It seems that another study could be done in the future using only your GPS collar data from both species and the justification would be similar to the authors by stating "......none of these studies assessed resource selection via a step selection approach..." or similar reasoning. Is there a particular reason the authors chose RSF over more detailed methods using step selection or individual location-specific approaches with only the GPS datasets? Do the authors feel their study design and methods on several study areas is more reliable and beneficial than on a study design using only GPS-collared predators?

CJP: On of the major goals of our study was to assess the generalizability of our predictions of predator space-use to other regions in Montana. Outside of the Garnet Range, the only other mountain lion telemetry data we had available was VHF-sourced. These VHF data had low temporal resolution, therefore, precluding the assessment of step selection function, which require location data sampled at a fine temporal resolution. Therefore, we chose to use RSFs for mountain lions and wolves, and wanted to keep our methodology consistent between predator species. At the end of the introduction, we added: 

“To assess the generality of our RSFs, we applied each study area-specific RSF to out-of-sample GPS telemetry data from separate study areas for wolves and assessed their predictive performance. Very high frequency (VHF) telemetry datasets were the only out-of-sample telemetry data available for mountain lions in northwest Montana (20), and consisted of low sample sizes relative to GPS datasets. This precluded us from assessing the generality of more detailed models like step selection functions because they require animal locations sampled at finer temporal scales than standard RSFs (55). However, VHF data enabled us to test the generality of mountain lion RSFs, and we used RSFs for both mountain lions and wolves to keep our methods consistent between species." 

We added the reference to Thurfjell et al. (2014) to the reference list to back the need for fine temporal resolution in SSFs.

We added the reference to Krawchuk et al. (2014) to back the claim that wolves can affect mountain lion habitat selection.

Line 154: remove plural for "lion" but add plural for "area".

CJP: Done

Line 158: Change "The Garnets" to the "Garnett Range" which is should be referred to as throughout for consistency (also Line 163).

CJP: Done

Line 160: Remove "Garnet" from mountain lions considering this is mentioned in the paragraph describing the Garnett Range.

CJP: Done

Lines 223-226: The author identify the "MFWP's biomedical protocol for free-ranging mountain lions" and wolves but only report University IACUC #A3327-01 for mountain lions in the Whitefish Range. Were any other Institutional Animal Care and Use Protocols conducted for lions or wolves for any of the other studies?

CJP: We were unable to locate IACUC protocol records for mountain lion data from the Rocky Mountain Front. Detailed capture and handling protocols for mountain lions on the Rocky Mountain Front are outlined in:

 Williams, J. S. (1992). Ecology of mountain lions in the Sun River area of northern Montana by. Montana State University.

For wolves, I tracked down the IACUC protocol: (AUP # 070–17). I added this to the manuscript.

---

## [Editor Report · Decision Letter 2]

5 Jul 2021

Habitat selection by wolves and mountain lions during summer in western Montana

PONE-D-21-03244R2

Dear Dr. Dr. Peterson,

We’re pleased to inform you that your manuscript has been judged scientifically suitable for publication and will be formally accepted for publication once it meets all outstanding technical requirements.

Kind regards,

W. David Walter, Ph.D.

Academic Editor

PLOS ONE
---

## [Editor Report · Acceptance letter]

13 Jul 2021

PONE-D-21-03244R2 

Habitat selection by wolves and mountain lions during summer in western Montana 

Dear Dr. Peterson:

I'm pleased to inform you that your manuscript has been deemed suitable for publication in PLOS ONE. Congratulations! Your manuscript is now with our production department. 

Kind regards, 

on behalf of

Dr. W. David Walter 

Academic Editor

PLOS ONE